# The AP-2 complex has a specialized clathrin-independent role in apical endocytosis and polar growth in fungi

Olga Martzoukou[1], Sotiris Amillis[1], Amalia Zervakou[1], Savvas Christoforidis[2,3], George Diallinas[1]*

[1]Department of Biology, National and Kapodistrian University of Athens, Athens, Greece; [2]Institute of Molecular Biology and Biotechnology-Biomedical Research, Foundation for Research and Technology, Ioannina, Greece; [3]Laboratory of Biological Chemistry, Department of Medicine, School of Health Sciences, University of Ioannina, Ioannina, Greece

**Abstract** Filamentous fungi provide excellent systems for investigating the role of the AP-2 complex in polar growth. Using *Aspergillus nidulans,* we show that AP-2 has a clathrin-independent essential role in polarity maintenance and growth. This is in line with a sequence analysis showing that the AP-2 $\beta$ subunit ($\beta$2) of higher fungi lacks a clathrin-binding domain, and experiments showing that AP-2 does not co-localize with clathrin. We provide genetic and cellular evidence that AP-2 interacts with endocytic markers SlaB[End4] and SagA[End3] and the lipid flippases DnfA and DnfB in the sub-apical collar region of hyphae. The role of AP-2 in the maintenance of proper apical membrane lipid and cell wall composition is further supported by its functional interaction with BasA (sphingolipid biosynthesis) and StoA (apical sterol-rich membrane domains), and its essentiality in polar deposition of chitin. Our findings support that the AP-2 complex of dikarya has acquired, in the course of evolution, a specialized clathrin-independent function necessary for fungal polar growth.

*For correspondence: diallina@biol.uoa.gr

**Competing interests:** The authors declare that no competing interests exist.

## Introduction

The five distinct Adaptor Protein (AP or adaptin) complexes are heterotetrameric adaptors that recruit membrane cargoes and coat proteins during vesicle formation at various subcellular locations for membrane trafficking in eukaryotes (*Nakatsu and Ohno, 2003*; *Robinson, 2004*, *2015*). AP-1 is necessary for the formation of clathrin-coated vesicles that traffic between the *trans*-Golgi network (TGN) and early endosomes. AP-2 is involved in the formation of clathrin-coated endocytic vesicles from the plasma membrane (PM). AP-3 is involved in, apparently, clathrin-independent vesicle formation in the Golgi for traffic to endosomes and lysosomes or vacuoles. AP-4 and AP-5 seem to have more specialized roles in vesicle transport of specific cargoes from the TGN to endosomes and/or the cell surface (*Hirst et al., 2013*). All AP complexes are composed of two different subunits of high molecular weight ~100 kDa (called $\alpha$, $\beta$ or $\gamma$ adaptins), one subunit of medium size ($\mu$, 47–50 kDa), and one low-molecular-weight subunit ($\sigma$, 17–19 kDa). The five AP complexes have been shown to have a wide eukaryote distribution, supporting that all were present in the Last Eukaryote Common Ancestor (LECA) (*Barlow et al., 2014*). Noticeably, however, each AP complex, except AP-1, has been lost in various lineages. Fungi possess homologues of all AP-1, AP-2 and AP-3 subunits, but the great majority of them, have lost AP-4 and AP-5. This loss concerns all Dikarya (Ascomycetes and Basidiomycetes) and most primitive fungi. AP-4 or AP-5 subunits are however present in the glomeromycete *Rhizophagus irregularis* and chytrid fungi *Spizellomyces punctatus* and

*Batrachochytrium dendrobatidis*, as well as, the sister group of all fungi, *Fonticula alba*. Notably, all eukaryotes studied possess clathrin, except for the Microsporidia intracellular parasites. Interestingly, the microsporidian *Encephalitozoon cuniculi* maintains AP subunits without having clathrin (*Barlow et al., 2014*), suggesting an unknown clathrin-independent role of AP complexes.

The role of the AP-2 complex in clathrin-mediated endocytosis is well established in mammals. By contrast, its role in endocytosis in unicellular eukaryotes, such as the yeasts *Saccharomyces cerevisiae*, *Schizosaccharomyces pombe* and *Candida albicans* has been questioned. An early systematic biochemical and genetic characterization of AP complexes in *S. cerevisiae* provided evidence that AP complexes are dispensable for clathrin function (*Yeung et al., 1999*). Similarly, *Huang et al. (1999)* have shown that AP deletion strains did not display the phenotypes associated with clathrin deficiency, including slowed growth and endocytosis, defective late Golgi protein retention and impaired cytosol to vacuole/autophagy function. Subsequently however, AP-2 has been localized to endocytic sites that are associated with clathrin, and have a cargo-specific function in killer toxin K28 endocytosis, making it likely that AP-2 functions with clathrin (*Carroll et al., 2009*, *2012*). Furthermore, AP-2 has been shown to be critical for hyphal growth in *C. albicans* and polarized cell responses in *S. cerevisiae*, and in particular to be necessary for relocalization of the cell wall stress sensor Mid2 to the tip of a mating projection following pheromone addition (*Chapa-y-Lazo et al., 2014*). Additionally, an interaction between the mu ($\mu$) subunit of AP-2 and the cell wall integrity pathway component Pkc1 was reported to affect recruitment of the AP-2 complex to endocytic sites (*Chapa-y-Lazo and Ayscough, 2014*). In another systematic screen to identify proteins required for cargo internalization, clathrin was shown to have a critical role for synaptobrevin homologue Snc1 endocytosis, but the role of AP-2 was not investigated (*Burston et al., 2009*). In a more recent study it has been shown that clathrin contributes to the regularity of vesicle scission and vesicle size, but is not required for elongating or shaping the endocytic membrane invagination, but again the role of AP-2 was not studied (*Kukulski et al., 2016*). Finally, in *S. pombe*, the AP-2 $\alpha$ component (Apl3p) was shown to interact physically with clathrin light chain (Clc1p), and genetically with both Clc1p and the endocytic factor Sla2p and additionally *apl3Δ* null mutants showed altered dynamics of endocytic sites associated with abnormal cell wall synthesis and morphogenesis (*de León et al., 2016*; *de León and Valdivieso, 2016*).

The apparently moderate role of AP-2 for growth in yeasts might reflect the fact that endocytosis in general is not essential for growth in unicellular fungi. In contrast to yeasts, the extreme polar growth of filamentous fungi, taking place by apical extension, is absolutely dependent on efficient and continuous endocytosis. Seminal studies on endocytosis and its relationship to growth have been performed in the basidiomycete *Ustilago maydis* (*Steinberg, 2014*) and the ascomycete *Aspergillus nidulans* (*Peñalva, 2010*), two emerging cell biology model systems.

Two aspects of endocytosis have been studied in detail in *A. nidulans*. The first relates to the essential role of apical endocytosis in hyphal tube extension and polar growth (*Araujo-Bazán et al., 2008*; *Taheri-Talesh et al., 2008*; *Hervas-Aguilar and Penalva, 2010*; *Peñalva, 2010*). It has been rigorously shown that polar growth maintenance is based on the spatial and functional coupling of rapid and continuous apical secretion and endocytosis in the hyphal tip region. Apically localized cargoes, as exemplified by the v-SNARE synaptobrevin homologue SynA, are internalized from a 'collar' region immediately below the hyphal tip and sorted in early endosomes (EEs). EEs can mature into Late Endosomes (LE) or Multivesicular Bodies (MVB) and be sorted into vacuoles, or recycle back, apparently via the *trans*-Golgi network (TGN) onto the PM. These post-endocytic trafficking processes taking place at the hyphal tips seem to be essential for the normal rate of secretion and proper growth. SlaB[End4] and actin polymerization are major factors for apical endocytosis.

A second aspect of endocytosis in *A. nidulans* is related to its role in PM transporter down-regulation in response to physiological or stress signals (*Gournas et al., 2010*; *Karachaliou et al., 2013*). Unlike endocytosis related to apical cargoes taking place mostly at hyphal tips, transporter internalization takes place homogenously all along the hyphal membrane and necessitates cargo ubiquitination. Transporter ubiquitination requires a group of cargo-specific $\alpha$-arrestin like proteins, acting as adaptors of the HECT-type ubiquitin ligase HulA[Rsp5]. Transporter ubiquitination is followed by internalization and sorting into EEs and the MVB/vacuolar pathway for terminal degradation. Transporter internalization, similar to apical endocytosis, requires actin polymerization and functional endocytic factors SagA[End3] or SlaB[End4].

Curiously, the role of clathrin or AP-2 in apical or transporter endocytosis had not, until very recently, been investigated in *A. nidulans,* or other filamentous fungi. An exception is a very recent report that studied clathrin dynamics in *A. nidulans*, following basically the subcellular localization and role in growth, of the clathrin heavy chain, ClaH (*Schultzhaus et al., 2017b*). This report showed that ClaH is localized mostly in late (*trans*) Golgi and secondarily to tentative endocytic sites, and although absolutely essential for growth, it does not seem to be involved in endocytosis. The later conclusion was, however, solely supported by studies concerning the internalization of the FM4-64 lipophilic marker, rather than specific endocytic cargoes. This report did not investigate the role of AP-2, or any AP complex, in relation to clathrin or to *A. nidulans* growth.

In this work, we investigate the role of the AP-2 complex and clathrin in polar growth and endocytosis in *A. nidulans*. We present evidence that the roles of AP-2 and clathrin are distinct, apparently due to an evolutionary truncation of the clathrin-binding domain in the β subunit of AP-2 in all higher fungi. We further provide evidence that AP-2 has a specialized, clathrin-independent role related to maintaining proper apical lipid and cell wall composition and thus promoting polar growth.

## Results

### Fungal AP complexes possess evolutionary truncated β subunits

A recent evolutionary analysis studied the distribution and origin of all AP complex subunits in fungi (*Barlow et al., 2014*). The conclusion from this thorough study was that all fungi possess AP-1, AP-2 or AP-3 subunits, but most, and apparently all Dikarya, have lost AP-4 or AP-5. Given our interest on the role of AP1-3 complexes in membrane traffic and particularly of AP-2 in endocytosis in *A. nidulans*, we carried out a similar analysis with particular emphasis on *Aspergillus* species. *Figure 1—figure supplement 1* shows that all Aspergilli possess highly similar homologous AP1-3 subunits, which can be unambiguously assigned into the distinct AP complexes. *Aspergillus* AP subunits are also phylogenetically highly related to homologues from all major fungal groups, as well as, selected model organisms (*Figure 1—figure supplement 2*). Importantly, AP subunits of a given complex are more similar to those of the orthologous complex in distantly related species, than to subunits of other AP complexes of the same species, in line with the acquisition of different functional roles of AP-1,–2 or −3 complexes in cargo membrane traffic.

A major and somehow surprising finding concerning the β subunits of not only *Aspergillus* species, but most fungi (all dikarya) came from the sequence analyses we performed. We observed that the β subunits of all three AP complexes do not contain the C-terminal part, of approximately 332 amino acid residues, which however is present in some primitive fungi, such as the chytrids *Rozella allomycis*, *Allomyces macrogynus Batrachochytrium dendrobatidis*, *Spizellomycetales punctatus* and *Macrophomina phaseolina* and in the fungal sister group *Fonticula alba* (all sharing 35–40% identity with human β subunits) (*Figure 1A*). The C-terminal domain of the β subunit is conserved in several Amoebozoa (25–36% identity), Rhizaria (31% identity), Alveolata (29% identity in *Plasmodium* or *Toxoplasma* species), Choanoflagellates (34–44% identity), Mesomycetozoea (44–50% identity), Algae (30% identity), and in the great majority of plants (38% identity) and Metazoa (58–60% identity) (*Figure 1B* and data not shown). Some prominent losses, except fungi, were identified in *Trypanosoma* and *Leishmania* species and in Excavata. Importantly, the lost part corresponds to an essential part of β appendage, which includes two C-terminal domains (pfam02883 and pfam09066) involved in clathrin binding and polymerization, and consequently necessary for the translocation of endocytic accessory proteins to a membrane bud site (*Mousavi et al., 2004*; *Lemmon and Traub, 2012*). A hydrophobic patch present in the lost domain also binds to a subset of DΦ[F/W] motif-containing proteins that are bound by the alpha-adaptin appendage domain (epsin, AP180) (*Lemmon and Traub, 2012*). Thus, fungal AP complexes might be incapable of interacting, at least via a 'canonical' association, with clathrin. Microsporidia, which are probably the only known eukaryotes lacking clathrin, also lack the clathrin binding domains in their AP complexes (*Barlow et al., 2014*). A major prediction from these observations is that, in fungi, clathrin-dependent processes, such as endocytosis and vesicle budding, might be AP-complex independent, while any role of the AP complexes might in turn be clathrin-independent. Obviously this prediction does not exclude that AP-2 or AP-1 complexes might still have the potential to interact with clathrin through unknown binding sequences in the β subunit or domains in other subunits, as for example reported for AP-1

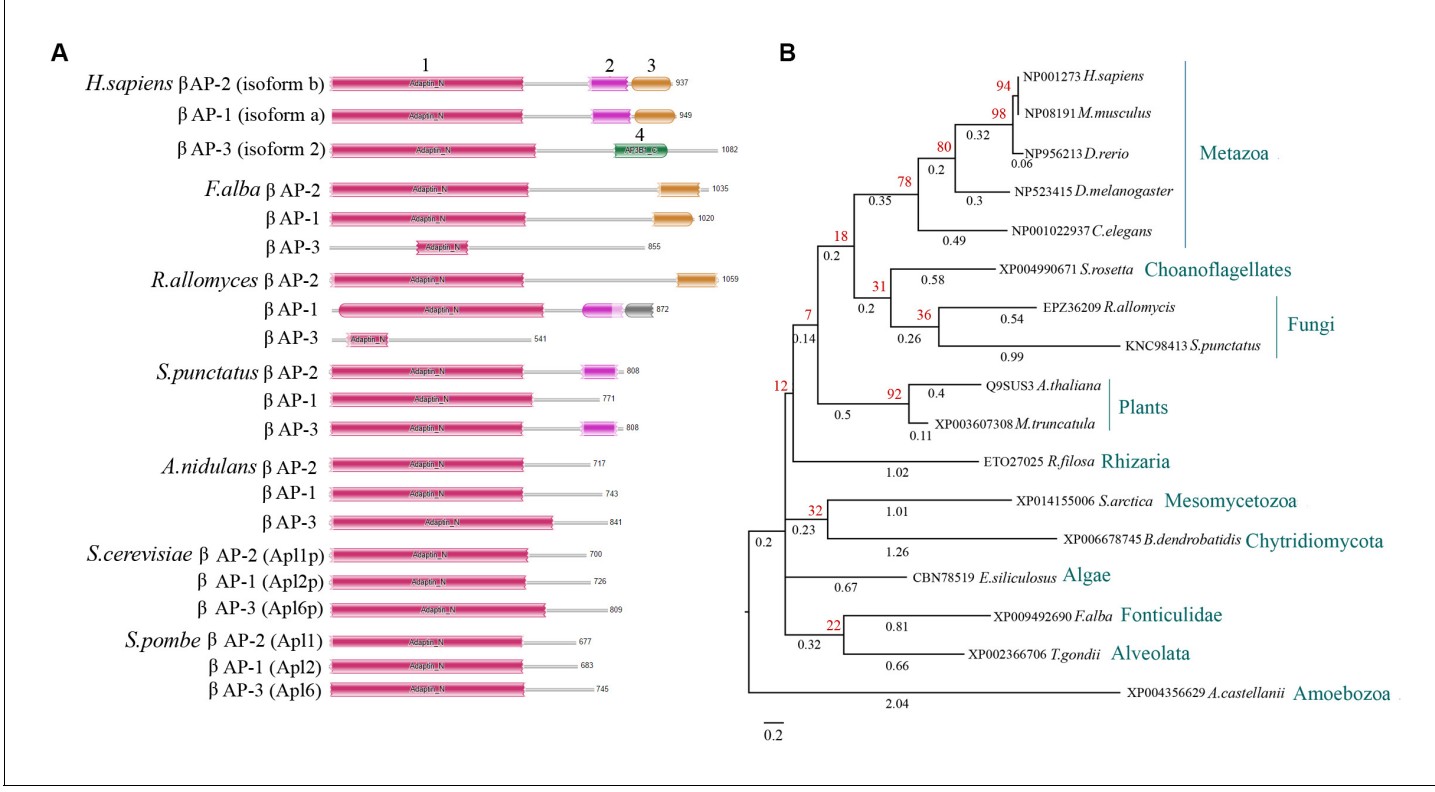

**Figure 1.** The β subunit of fungal AP complexes lacks clathrin-binding domains. (**A**) Cartoon depicting the absence in the β subunits of AP complexes of higher fungi (*A. nidulans*, *S. cerevisiae* or *S. pombe*) of a C-terminal region that includes putative clathrin binding domains. The cartoon includes the human AP1-3 β subunits as examples of canonical AP complexes, as well as, selected primitive fungi (*Rozella allomycis* and *Spizellomycetales punctatus*) and *F. alba*, as examples of lower eukaryotes conserving degenerate versions of putative clathrin binding domains. One signifies the N-terminal adaptin domain (pfam01602, ~534 amino acids) common in β subunits of all AP complexes. Two is the α-adaptin C2 domain (pfam02883, ~111 amino acids) present in the β subunits of AP-2 and AP-1. Three is the β2-adaptin appendage (pfam09066, ~112 amino acid residues) present in the β subunits of AP-2 and AP-1, required for binding to clathrin. Four is the so-called clathrin adaptor protein complex β1 subunit domain, found in AP-3 complexes, probably required for Golgi association (pfam14796, ~148 amino acids). (**B**) Phylogenetic relationships of the C-terminal region of the β subunit of AP-2 that includes clathrin-binding domains. The tree includes selected organisms representing major taxonomic groups, from the amoeba *A. castellanii* to metazoa such as *C. elegans* and *H. sapiens*. The tree was reconstructed with the maximum-likelihood method and bootstrap-method testing (shown in red). The branch scale used was 0.2 and the branch lengths (shown in black) reflect the expected number of substitutions per site.

The following figure supplements are available for figure 1:

**Figure supplement 1.** Phylogenetics of the four subunits of AP1, AP-2 and AP-3 of Aspergilli with three model organisms used as out-groups.

**Figure supplement 2.** Phylogenetics of the four subunits of AP-1, AP-2 and AP-3 of major fungal groups with out-groups from model organisms.

in *S. cerevisiae* (**Yeung and Payne, 2001**). However, subsequent experimental evidence presented herein strongly support a clathrin-independent role of AP-2 in *A. nidulans* polar growth.

## AP-1 and AP-2, but not AP-3, complexes are critical for polarity maintenance and *A. nidulans* growth

To investigate the role of AP complexes in cargo membrane traffic and endocytosis in *A. nidulans* we constructed three null mutants, each carrying a deletion of the gene encoding the σ subunit of AP-1 (AN7682; *apaS*), AP-2 (AN0722; *apbS*) or AP-3 (AN5519; *apcS*) (http://www.aspergillusge-nome.org/ and **Osmani and Goldman, 2008**). Our primary interest was to examine the role of AP-2 in endocytosis, which however also necessitated the knock-out of the σ subunits of AP-1 and AP-3 in addition to that of AP-2, for excluding any possible functional complementation due to the similarity

of the three σ subunits. Genetic disruption of the σ subunits, similarly to disruption of any of the four AP subunits, has been reported to inactivate the function of the full adaptor complex (*Robinson, 2004, 2015*). Isogenic null mutants were constructed and named *thiA*$_p$-*ap1*$^\sigma$, *ap2*$^\sigma\Delta$ or *ap3*$^\sigma\Delta$ (see Materials and methods). For the *ap1*$^\sigma$ gene, given its knock-out proved lethal (results not shown), we constructed a conditional knock-down strain using the thiamine-repressible promoter, *thiA*$_p$ (*Apostolaki et al., 2012*). The σ subunit null mutant strains were analyzed phenotypically and microscopically.

Knocking-down *ap1*$^\sigma$ or deleting *ap2*$^\sigma$ severely affected growth rate and colony morphology, whereas the *ap3*$^\sigma\Delta$ mutant exhibited only a very minor delay in growth rate (inserts in *Figure 2A*). The double mutant strains *ap2*$^\sigma\Delta$ *thiA*$_p$-*ap1*$^\sigma$ and *ap2*$^\sigma\Delta$*ap3*$^\sigma\Delta$, constructed via standard genetic crossing, had growth rates similar to the single *thiA*$_p$-*ap1*$^\sigma$ or *ap2*$^\sigma\Delta$ mutant, showing that the defect in *ap2*$^\sigma\Delta$ is independent of AP-1 or AP-3 complexes. All mutants tested remained sensitive to selected antifungals, such as 5-flourouracil, 5-fluorocytosine or 8-azaguanine, similar to wild-type isogenic controls, suggesting that the expression and/or turnover of transporters specific for these antifungals is not affected (not shown).

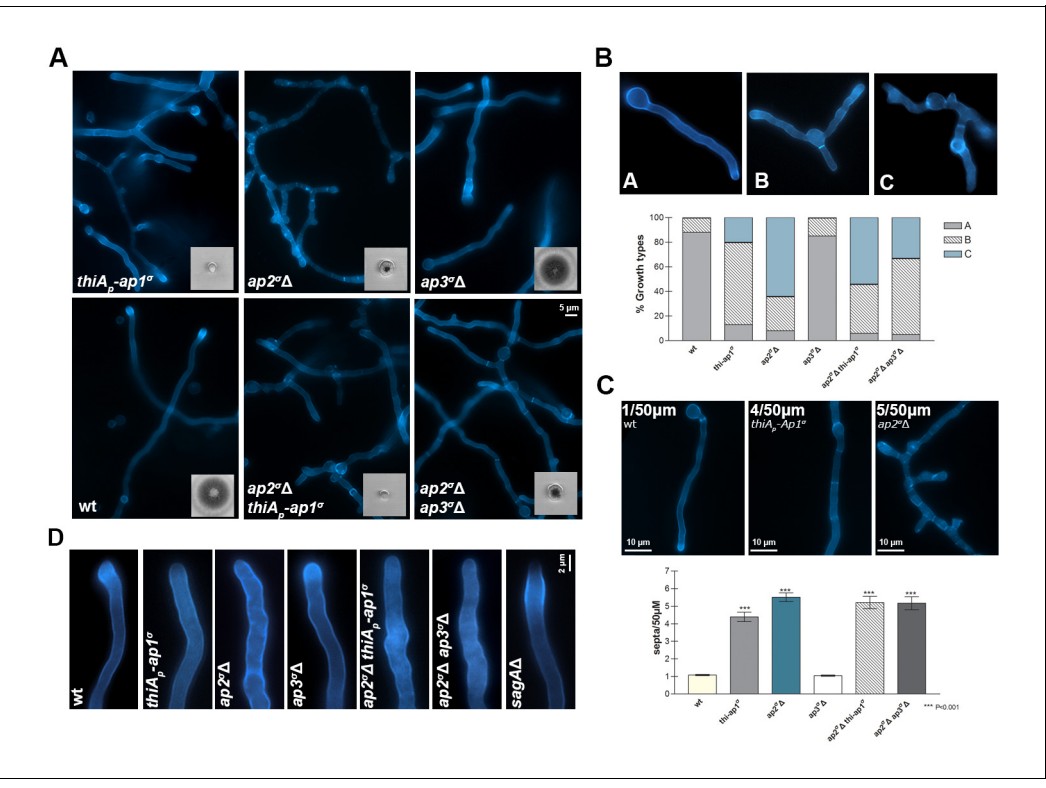

**Figure 2.** AP-1 and AP-2, but not AP-3, complexes are critical for *A. nidulans* growth. (**A**) Colony growth (bottom left inserts) and microscopic morphology (20 hr hyphal cells stained with calcofluor) of isogenic *thiA*$_p$-*ap1*$^\sigma$, *ap2*$^\sigma\Delta$, *ap3*$^\sigma\Delta$, *ap2*$^\sigma\Delta$ *thiA*$_p$-*ap1*$^\sigma$ and *ap2*$^\sigma\Delta$ *ap3*$^\sigma\Delta$ mutant strains, compared to wild-type (wt). Biological/Technical replicates: 4/25, 3/12, 3/15, 3/10, 2/10, 2/10 for wild-type and mutant strains, respectively. For the definition of the two categories of replicates see Materials and methods. (**B**) Representative types of morphological phenotypes related to unipolar or multipolar germination (upper panel) and relative quantitative analysis of n = 100 hyphae of wild-type and n = 84, 58, 60, 48 and 31 hyphae of *thiA*$_p$-*ap1*$^\sigma$, *ap2*$^\sigma\Delta$, *ap3*$^\sigma\Delta$, *ap2*$^\sigma\Delta$ *thiA*$_p$-*ap1*$^\sigma$ and *ap2*$^\sigma\Delta$ *ap3*$^\sigma\Delta$, respectively (lower panel). Replicates as in (**A**). (**C**) Representative types of septa formation (upper panel) and relative quantitative analysis of n = 32 hyphae of wild-type and n = 32, 23, 32, 32 and 27 hyphae of *thiA*$_p$-*ap1*$^\sigma$, *ap2*$^\sigma\Delta$, *ap3*$^\sigma\Delta$, *ap2*$^\sigma\Delta$ *thiA*$_p$-*ap1*$^\sigma$ and *ap2*$^\sigma\Delta$ *ap3*$^\sigma\Delta$ respectively (lower panel). Replicates as in (**A**). p<0.001 for *thiA*$_p$-*ap1*$^\sigma$, *ap2*$^\sigma\Delta$, *ap2*$^\sigma\Delta$ *thiA*$_p$-*ap1*$^\sigma$ and *ap2*$^\sigma\Delta$ *ap3*$^\sigma\Delta$, compared to wt. See Materials and methods for statistical analysis methods and statistical tests used. (**D**) Calcofluor deposition at the tip of growing hyphae in AP mutants and wild-type. A standard endocytic mutant, *sagA*$\Delta$, showing reduced calcofluor staining at the tip is included for comparison. For experimental details see Materials and methods. Replicates as in (**A**).

We examined the microscopic morphology of the mutants constructed. $thiA_p$-$ap1^\sigma$ and $ap2^\sigma\Delta$, but not $ap3^\sigma\Delta$, showed altered hyphal morphology, characterized by anomalous polar tube hyper-branching and shorter coenocytic compartments, that is, reduced septa distances (*Figure 2A*). This effect was confirmed by quantification of the distinct microscopic morphological phenotypes and septa frequencies, as shown in *Figure 2B and C*. We also tested calcofluor deposition at the apex of polar tips in AP complex mutants, relatively to wild-type. Calcofluor stains chitin and the absence of intense calcofluor-staining at the tip indicates abnormal cell wall synthesis, related to problematic recycling of chitin synthases. This has been observed in mutants blocked in apical endocytosis (*Higuchi et al., 2009*). Calcofluor localization was totally depolarized in $thiA_p$-$ap1^\sigma$ and $ap2^\sigma\Delta$, similar to what was observed in the endocytic mutant SagA (*Figure 2D*). Thus, the overall picture obtained with $thiA_p$-$ap1^\sigma$ and $ap2^\sigma\Delta$ mutants was that of defective maintenance of polarity (*Momany, 2002*; *Steinberg, 2014*).

## AP complexes are dispensable for membrane traffic and endocytosis of transporters

We investigated the role of the AP complexes on membrane cargo traffic and endocytosis by following the subcellular localization of a GFP-tagged transporter, namely UapA. UapA is an extensively studied uric acid-xanthine/$H^+$ symporter, which has been used as a model cargo for studying the regulation of exocytosis and endocytosis of polytopic membrane proteins in *A. nidulans* (*Pantazopoulou et al., 2007*; *Gournas et al., 2010*; *Karachaliou et al., 2013*). After synthesis and translocation in the ER, fully functional GFP-tagged UapA molecules follow a vesicular secretion pathway to localize homogenously and stably in the PM of hyphal cells. However, UapA molecules undergo ubiquitylation-dependent endocytosis and vacuolar degradation in response to substrate excess or to the presence of rich nitrogen sources ($NH_4^+$ or glutamine). This picture remained unaffected when we followed the subcellular localization of UapA-GFP in isogenic $thiA_p$-$ap1^\sigma$, $ap2^\sigma\Delta$ or $ap3^\sigma\Delta$ mutant backgrounds (*Figure 3A*). The same picture is obtained under conditions of de novo or continuous UapA-GFP synthesis, or when signals triggering endocytosis are imposed after repression of UapA-GFP synthesis (data not shown). The non-essentiality of the AP-1 and AP-3 complexes in UapA traffic was somehow expected, because transporter exocytosis occurs by direct fusion of post-Golgi vesicles with the PM, and seemingly does not involve sorting from the Golgi to the endosomal compartment. However, the dispensability of AP-2, an adaptor involved in the formation of endocytic vesicles in other systems, for UapA-GFP endocytosis was somehow unexpected and contrasted the picture observed in standard endocytic mutants ($sagA\Delta$ or $thiA_p$-$slaB$), where UapA internalization from the PM was totally blocked (see right panels in *Figure 3A* and *Figure 3—figure supplement 1* for the construction of $thiA_p$-$slaB$). The non-involvement of AP-2 in UapA endocytosis was further confirmed by time-lapse experiments, performed in the $ap2^\sigma\Delta$ mutant strain and an isogenic wild-type control ($ap2^{\sigma+}$), which showed that UapA-GFP internalization initiates at 15 min and becomes more prominent 20–30 min after $NH_4^+$ addition in both strains (*Figure 3B*).

The non-involvement of AP-2 in UapA endocytosis raised issues worthy to be investigated. Firstly, we tested whether the endocytosis of other evolutionary, structurally and functionally distinct transporters is also AP-2 independent. *Figure 3C,E and F* show that this is indeed the case, as all six transporters tested are internalized with the same efficiency and rate, in both $ap2^\sigma\Delta$ and $ap2^{\sigma+}$ backgrounds, in response to endocytic signals (*Tavoularis et al., 2001*; *Vlanti and Diallinas, 2008*; *Apostolaki et al., 2009*; *Krypotou et al., 2015*). Noticeably, different transporters exhibit different stabilities and endocytic 'sensitivities', but as quantification confirmed (*Figure 3E and F*), in all cases AP-2 proved dispensable for their proper internalization rate and turnover. We also examined whether the genetic elimination of a different AP-2 subunit, other than σ2, also leads to similar results in respect to transporter endocytosis. We deleted the gene encoding the μ2 subunit of the AP-2 complex, and subsequently constructed $ap2^\sigma\Delta$ $ap2^\mu\Delta$ double mutants and strains expressing UapA-GFP in the $ap2^\mu\Delta$ background. The $ap2^\sigma\Delta$, $ap2^\mu\Delta$ and $ap2^\sigma\Delta$ $ap2^\mu\Delta$ mutants had identical phenotypes, characterized by loss of polarity maintenance and poor growth rate (*Figure 3—figure supplement 2*). UapA-GFP endocytosis operated normally in the $ap2^\mu\Delta$ background, similar to what was observed in the $ap2^\sigma\Delta$ or wild-type isogenic backgrounds (*Figure 3D*). Overall, these results confirmed that AP-2 is dispensable for transporter endocytosis.

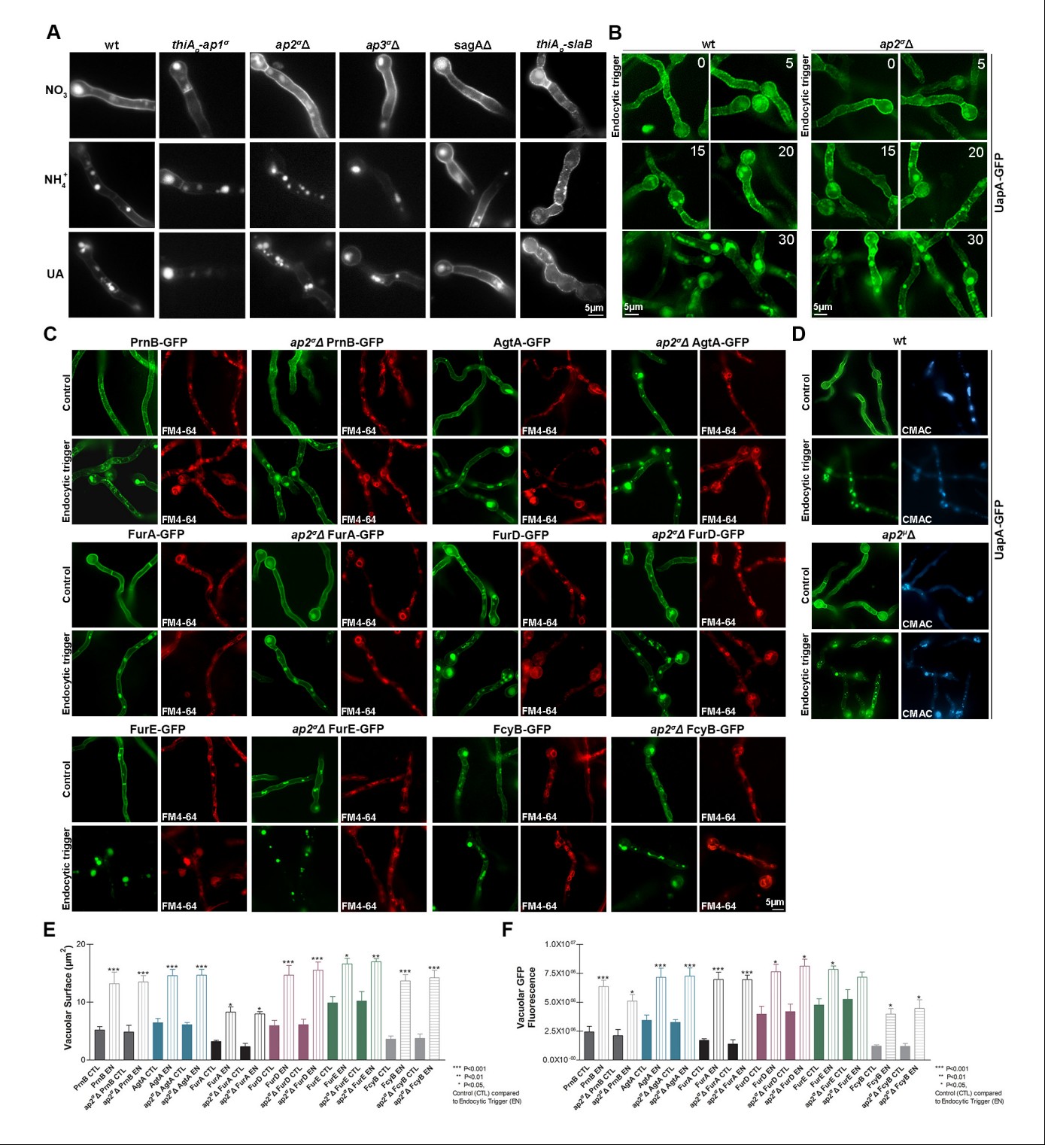

**Figure 3.** AP complexes are dispensable for transporter membrane traffic and endocytosis. All panels show Epifluorescence microscopy analyses of 18–20 hr growing hyphal cells in supplemented Minimal Media. (**A**) Subcellular localization of UapA-GFP under standard growing conditions ($NO_3^-$ as sole N source) and in response to endocytic signals (standard conditions followed by 2 hr addition of either $NH_4^+$ or uric acid, UA) in AP (*thiA_p-ap1^σ*, *ap2^σΔ*, *ap3^σΔ*) or endocytic mutants (*sagAΔ* and *thiA_p-slaB*), compared to isogenic wild-type. Biological/Technical replicates: 2/10. (**B**) Time course (min) of $NH_4^+$-elicited endocytosis of UapA-GFP in a *ap2^σΔ* genetic background. An identical picture is obtained in a wild-type (*ap2^σ+*) background (*Gournas et al., 2010*). Biological/Technical replicates: 2/10. (**C**) Subcellular localization of six GFP-tagged transporters belonging to distinct protein families (PrnB, AgtA, FurA, FurD, FurE and FcyB; for details see text) in response to endocytic trigger (2 hr $NH_4^+$). Staining with FM4-64 is included to
*Figure 3 continued on next page*

*Figure 3 continued*

show that in the presence of NH$_4^+$ all transporters are eventually sorted for degradation in the vacuoles, similarly to what was observed with UapA. Biological/Technical replicates: 2/20, 2/20, 2/15, 2/15, 2/15 and 3/10. (D) Relative subcellular localization of UapA-GFP in response to endocytic trigger (2 hr NH$_4^+$) in wild-type and *ap2$^H\Delta$* genetic backgrounds. CMAC staining highlights terminal sorting in the vacuoles in the presence of NH$_4^+$. Notice that UapA-GFP is normally endocytosed in *ap2$^H\Delta$*, similarly to *ap2$^r\Delta$* or the wild-type strain. Biological/Technical replicates: 2/12. See also ***Figure 3—figure supplement 2***. (E–F) Quantitative analysis of transporter endocytosis presented in (C) as depicted by measurements of vacuolar surface or vacuolar GFP fluorescence. n = 5 hyphae per condition (Control or Endocytosis). For the method of measurements, statistical analysis and other experimental details see Materials and methods. Replicates as in (C).

The following figure supplements are available for figure 3:

**Figure supplement 1.** Phenotypic characterization of a conditional null mutant of SlaB constructed using the *thiA$_p$* repressible promoter.

**Figure supplement 2.** Phenotypic characterization of conditional *ap2$^H\Delta$* null mutants constructed using standard reverse genetics and genetic crossing.

## Clathrin and AP-2 have distinct roles in *A. nidulans* growth and transporter endocytosis

We showed that AP-2 is essential for polarity maintenance, but dispensable for transporter trafficking and endocytosis. To investigate the role of clathrin in these processes, we constructed relevant null or conditional mutants of the heavy (ClaH) and light (ClaL) chains of clathrin. The ClaH null mutant (*claH*) was not viable as it could only be rescued in heterokaryotic transformants (*Osmani et al., 2006*). The ClaL null mutant (*claL*) was viable, but severely affected in growth (*Figure 4A*). The corresponding conditional mutants were based on the use of the highly repressible *thiA$_p$* promoter (*Apostolaki et al., 2012*). In the absence of thiamine (repressor), strains expressing *thiA$_p$-claH* or *thiA$_p$-claL* grew similar to wild-type isogenic control strains (not shown), whereas in the presence of thiamine *thiA$_p$-claH* did not grow to form colonies, while *thiA$_p$-claL* formed very small compact colonies, compatible with a severe growth defect (*Figure 4A*). *Schultzhaus et al. (2017b)*, recently reported very similar growth phenotypes. We further examined the thiamine-repressible mutants under the microscope (*Figure 4B*). The *thiA$_p$-claL* strain showed normal hyphae morphology and polar growth in the absence of thiamine, but prominent hypha tip swelling under *ab initio* repression of ClaL, that is, in the continuous presence of thiamine, especially at 37°C. This phenotype, which was confirmed by quantification (*Figure 4C–E*), might be associated with genetic blocks in apical endocytosis of specific cargoes. Knock-down of ClaL did not however modify the polar depositioning of calcofluor staining, which suggests normal chitin synthesis (*Higuchi et al., 2009*; *Takeshita et al., 2012*), and neither affected polarity maintenance. The *thiA$_p$-claH* strain also showed normal hyphae morphology and polar growth in the absence of thiamine, but contrastingly, hyphae morphology was severely affected in the continuous presence of thiamine, becoming flattened and often showing swelling of tips or conidiospore heads, or becoming thinner and smaller in overall size, an indication of progressive lethality (*Figure 4B and F*). Still however, polarity maintenance was preserved when ClaH was repressed. Thus, in no case the phenotypes resulting from the absence of clathrin, either ClaH or ClaL, resembled those obtained in the absence of AP-2.

We also asked whether clathrin has a role in transporter endocytosis. Knockdown of ClaL expression blocked ammonium-elicited endocytosis of the UapA transporter, but knockdown of ClaH expression showed a more complex picture (*Figure 4G*). More specifically, in the latter case, when thiamine was added from the beginning of growth, UapA-GFP showed a rather diffuse and cytoplasmic appearance, instead of the normal cortical marking of the PM (notice the magnified area in the relevant insert in *Figure 4G*). This picture was obtained independently from the presence or absence of the endocytic trigger. Under the light of the recent publication of *Schultzhaus et al. (2017b)*, which showed that ClaH is principally localized in the late Golgi, but also based on our own independent results on the effect of ClaH on other apical markers, described later in this work, it seems that *ab initio* knockdown of ClaH severely affects secretion of cargoes, including transporters. This leads to apparent Golgi and endosome collapse and progressive lethality, which could easily explain why we detect UapA-GFP in cytosolic structures, rather than in the PM. In addition, a block in transporter secretion would also create a cellular stress signal resulting in further internalization and turnover of transporters.

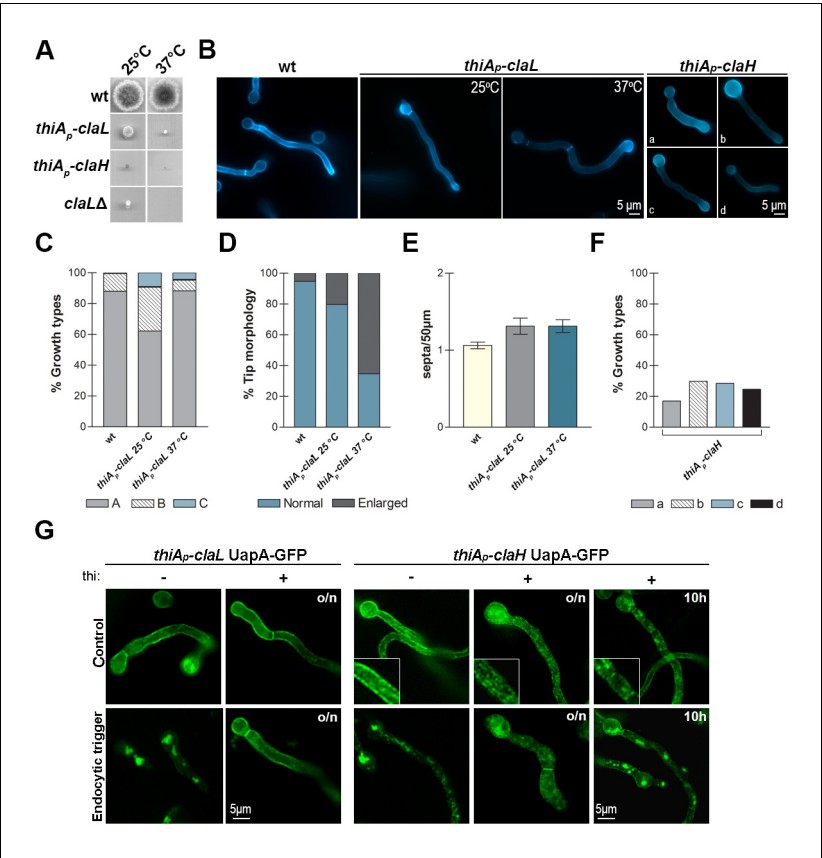

**Figure 4.** Clathrin is essential for growth and transporter secretion or/and endocytosis. (**A, B**) Colony growth phenotypes and microscopic morphology (hyphal cells stained with calcofluor) of ClaL knock-out (*claLΔ*) or conditionally knocked-down (*thiAₚ-claL* and *thiAₚ-claH*) mutants. Representative types of morphological phenotypes (a-d) of *thiAₚ-claH* are shown (B, right panel). For details see text. Biological/Technical replicates in (**B**): 4/25, 2/20 and 2/100, for the three strains respectively. Unless otherwise stated, thiamine was added *ab initio* (16 hr) at a final concentration of 10 μg ml$^{-1}$. (**C–E**) Quantitative analysis of growth phenotypes (categorized as in *Figure 2* in **A**,**B** or **C**), tip morphology and number of septa, in *thiAₚ-claL* (thiamine-repressed) and an isogenic wild-type control (wt). (**C**) Analysis of n = 100 hyphae of wild-type and n = 45, 94 hyphae of *thiAₚ-claL* at 25°C and 37°C respectively. (**D**) Analysis of n = 95, 69, 95 hyphal tips of wt, *thiAₚ-claL* at 25°C and 37°C respectively. (**E**) Analysis of n = 32 hyphae of wt and knock-down strains. Replicates as in (**B**). (**F**) Relative quantitative analysis of growth types shown in (**B**), of n = 200 hyphae of *thiAₚ-claH* (thiamine-repressed). Replicates as in (**B**). (**G**) Epifluorescence microscopy showing the relative subcellular localization of UapA-GFP under control or endocytic conditions (2 hr $NH_4^+$) in isogenic wild-type and *thiAₚ-claL* or *thiAₚ-claH* genetic backgrounds. Notice that repression of *claL* expression (o/n thiamine) blocks UapA-GFP endocytic turnover, *ab initio* repression of *claH* expression (o/n thiamine) severely blocks UapA-secretion to the PM, whereas *claH* repression (10 hr thiamine) after pre-secretion of UapA-GFP into the PM (14 hr) leads to an apparent block in secretion, but a fraction of UapA-GFP still remains in the PM. For more explanations see the text. Biological/Technical replicates: 4/10, 3/15 for *thiAₚ-claL* UapA-GFP and *thiAₚ-claH* UapA-GFP, respectively.

The following figure supplements are available for figure 4:

**Figure supplement 1.** Western blot analysis of *thiAₚ*-claH-GFP.

**Figure supplement 2.** Time course of FM4-64 internalization in wild-type and mutant strains.

To allow secretion and proper localization of UapA-GFP to the PM to take place, and subsequently to check the role of ClaH in endocytosis, we decided to repress *claH* expression by adding thiamine after a period of pre-growth (14 hr) in the absence of thiamine. For this, we directly tested whether 5 or 10 hr repression of *claH via* the *thiA* promoter is sufficient for depleting ClaH, by performing and quantifying immunoblot assays using anti-GFP or anti-actin antibodies against total protein extracts from a strain expressing ClaH-GFP. Results shown in *Figure 4—figure supplement 1* confirmed that addition of thiamine for ≥10 hr dramatically reduced ClaH. In contrast, 5 hr addition of thiamine had no effect on ClaH steady state levels, apparently because the relevant polypeptide is quite stable. Based on these immunoblots, we examined the effect of ClaH on UapA-GFP endocytosis by using at least a 10 hr period of *claH* repression, after pre-secretion of UapA-GFP into the PM. In this case, a significant fraction of UapA-GFP remained PM-associated despite the appearance of UapA-specific cytoplasmic structures and increased vacuolar degradation, which are however expected outcomes resulting from a block in Golgi functioning and transporter secretion (*Figure 4G*, extreme right panel).

Overall, our results strongly suggest that although both chains of clathrin seem essential for transporter (UapA-GFP) endocytosis, they do have distinct roles in transporter secretion. ClaL is dispensable for Golgi functioning and transporter secretion towards the PM, while ClaH seems absolutely essential for Golgi functioning and cargo secretion. These results further support that clathrin and the AP-2 complex have distinct roles in respect to both cargo secretion and transporter endocytosis.

Notably, we also found that FM4-64 endocytosis is not affected by deletions of genes encoding AP-2 complex subunits, clathrin or the endocytic factors SagA and SlaB, suggesting still another pathway of endocytosis in *A. nidulans* (*Figure 4—figure supplement 2*).

## AP-2, but not clathrin, is essential for the polar localization of lipid flippases DnfA and DnfB

Guided by the fact that lack of a functional AP-2 complex (*ap2*$^σ$*Δ* or *ap2*$^μ$*Δ* mutants) leads to defective polarity maintenance and dramatically reduced growth, we tested whether AP-2 has a specific role in the polar localization of well-studied apical cargoes, such as the v-SNARE SynA (*Valdez-Taubas and Pelham, 2003*; *Pantazopoulou and Peñalva, 2011*) or the DnfA and DnfB flippases (*Schultzhaus et al., 2015*). These are the only presently known recycling cargoes of the endocytic collar shown rigorously to be involved in polar growth (*Peñalva, 2015*). In parallel, we examined the role of AP-2 in the polar localization of the endocytic markers, SlaB (*Araujo-Bazán et al., 2008*; *Hervas-Aguilar and Penalva, 2010*) and SagA (*Karachaliou et al., 2013*), or of AbpA, which marks the sites of actin polymerization (*Araujo-Bazán et al., 2008*). Appropriate strains were constructed by standard genetic crossing. *Figure 5A* shows that in the *ap2*$^σ$*Δ* genetic background there is total depolarization of DnfA and DnfB, as these proteins now mark the hyphal PM rather homogenously (see also *Figure 5E*, *Figure 5—figure supplement 1A*). In contrast, absence of a functional AP-2 did not detectably affect the polar localization of SagA, SlaB, SynA or AbpA. Quantification of polarized *versus* non-polarized localization of these markers confirmed our conclusions (*Figure 5B*). The same overall results were obtained when we used the *ap2*$^μ$*Δ* mutant rather than *ap2*$^σ$*Δ* (results not shown).

We tested the role of clathrin (ClaL) in the polar localization of DnfA, SlaB and SagA. *Figure 5C* shows that in a *thiA$_p$-claL* strain repressed *ab initio* for ClaL expression, all markers tested remain polarly localized, marking principally the collar region behind the tip (see also *Figure 5E*). We also tested the role of ClaH in DnfA or DnfB localization (*Figure 5D* and *Figure 5—figure supplement 1B*). In a *thiA$_p$-claH* genetic background, *ab initio* (that is, o/n) addition of thiamine led to cytoplasmic distribution of DnfA-GFP or DnfB-GFP, which in this case mark large, often bulbous, bodies and a membrane-like network (upper and lower left panels in *Figure 5D*), resembling the picture obtained when we followed UapA-GFP in the absence of ClaH (see *Figure 4G*), or when Golgi/endosomes collapse in response to the presence of Brefeldin A (BFA) (middle panels in *Figure 5D*). We further showed that addition of BFA in the culture medium of a strain expressing ClaH (de-repressed *thiA$_p$-claH*) leads to reversible de-polarization of DnfA-GFP (lower panel in *Figure 5—figure supplement 2*), but to irreversible de-polarization of DnfA-GFP and progressive apparent hyphae death (that is, hyphae cells become thinner) when ClaH expression is repressed (upper panel in *Figure 5—figure supplement 2*). This picture is in very good agreement with the reversibility of the negative effect of BFA on Golgi function, which has been rigorously studied in *Pantazopoulou and Penalva*

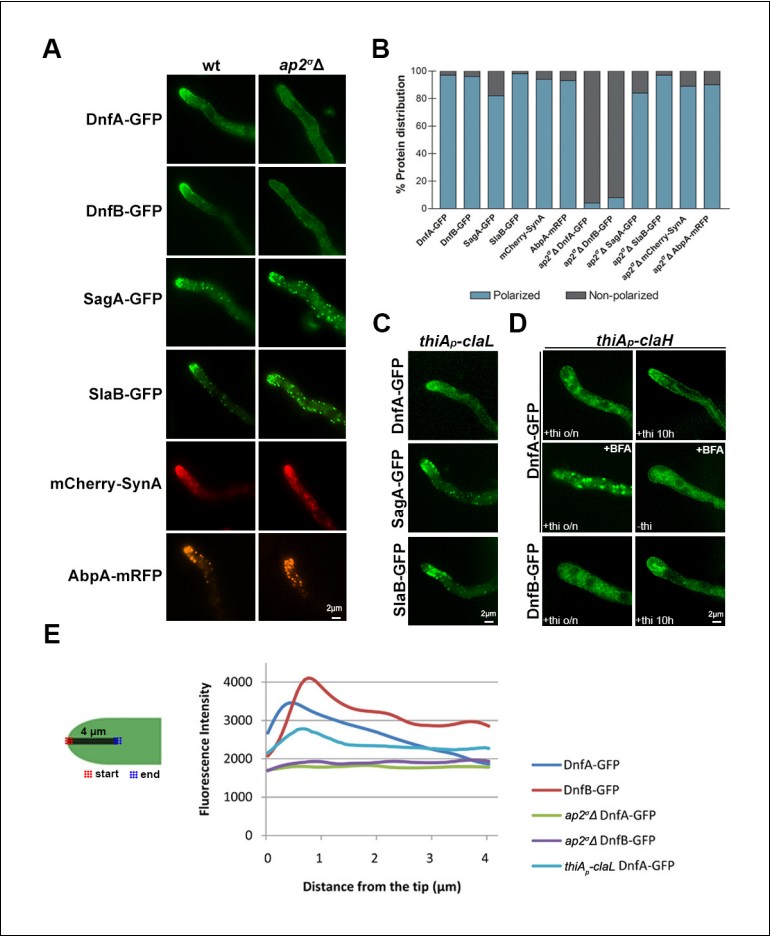

**Figure 5.** AP-2 is essential for the polar localization of lipid flippases DnfA and DnfB. (**A**) Epifluorescence microscopy showing the subcellular localization of apical markers DnfA, DnfB, SagA, SlaB, SynA and AbpA in wild-type and *ap2ᵅΔ* genetic backgrounds. DnfA and DnfB are lipid flippases, SagA and SlaB are factors involved in the formation of endocytic vesicles, AbpA is an actin-polymerization marker, and SynA is a v-SNARE marking the apical tip (for more details see the text). Notice that lack of a functional AP-2 complex leads to detectable depolarization of solely DnfA and DnfB. Representative phenotypes selected from 30–40 hyphae for wt and mutant strains. Biological replicates: 4. (**B**) Quantitative analysis of protein (apical marker) distribution (polarized *versus* non-polarized) of n = 58, 56, 51, 40, 32, 43 and 48, 50, 37, 60, 45, 61 hyphal tips of DnfA-GFP, DnfB-GFP, SagA-GFP, SlaB-GFP, mCherry-SynA, AbpA-mRFP in wild-type and *ap2ᵅΔ* genetic backgrounds, respectively. Replicates as in (**A**). (**C**) Epifluorescence microscopy of the subcellular localization of DnfA, SagA and SlaB in a *thiAₚ-claL* genetic background. Notice that ClaL repression (o/n thiamine) does not affect DnfA, SagA or SlaB polarization. Representative phenotypes selected from 20–30 hyphae for each strain. Biological replicates: 3. (**D**) Epifluorescence microscopy of the subcellular localization of DnfA or DnfB in a *thiAₚ-claH* genetic background. Notice that *ab initio* ClaH repression (o/n thiamine) affects DnfA or DnfB polarization (upper and lower left panels), apparently due to Golgi collapse (see text), while in samples repressed (10 hr thiamine) after a period of pre-growth (16 hr), a degree of polarization is retained (upper and lower right panels). The middle panel depicts DnfA localization in the presence of the Golgi inhibitor Brefeldin A (BFA), under ClaH repressed (o/n thiamine, 150 min BFA) or de-repressed conditions (25 min BFA). Notice the apparent block in DnfA-GFP secretion (also refer to *Figure 5—figure supplement 1* and the text for more details). Biological/Technical replicates: 3/50, 2/50 for *thiAₚ-claH* DnfA-GFP and *thiAₚ-claH* DnfB-GFP respectively. (**E**) Quantitative analysis of fluorescence intensity of DnfA-GFP or DnfB-GFP in wt, *ap2ᵅΔ* or *thiAₚ-claL* (thiamine-repressed), along 4 μm of hyphal tips. For details of fluorescence intensity measurements see Materials and methods.

The following figure supplements are available for figure 5:

**Figure supplement 1.** Subcellular localization of DnfA-GFP or DnfB-GFP in wild-type, *ap2ᵅΔ*, *thiAₚ*-claL or *thiAₚ*-claH isogenic backgrounds.

*Figure 5 continued on next page*

*Figure 5 continued*

**Figure supplement 2.** Time course of Brefeldin A effect on DnfA-GFP subcellular localization in a *thiA$_p$*-claH mutant.

*(2009)*. Importantly, our results show that normal Golgi secretion is necessary for maintaining the polar localization of DnfA-GFP.

When ClaH was depleted, by thiamine addition for 10 hr, following a period of pre-growth (16 hr) that allows a fraction of DnfA-GFP or DnfB-GFP to be secreted and polarly localized, a degree of polarization of both flippases was preserved in some hyphae, but in general the polar localization of DnfA-GFP or DnfB-GFP was disrupted (see upper and lower right panels in *Figure 5D* and also *Figure 5—figure supplement 1B*). This picture, however, did not constitute a surprise, given the effect of BFA in DnfA-GFP polar localization. It is rather compatible with the idea that a severe block in conventional cargo secretion, obtained either by BFA or by the depletion of ClaH, leads to an inability to *maintain* the apical localization of markers, such as DnfA, and apparently DnfB.

In summary, our results strongly suggest that the depletion of clathrin does not directly affect the pre-established polarization of flippases. While the dispensability of ClaL in apical cargo localization is clear, the non-essential role of ClaH on flippase polarization establishment is more difficult to formally confirm because depletion of ClaH affects Golgi function and secretion, and thus polarization maintenance. Our results are in agreement with results reported in *Schultzhaus et al. (2017a)*; *(2017b)*, which establish that ClaH is localized principally in late Golgi, only weakly in the apical collar region and is excluded from the hyphal tip. Overall, these results further showed that AP-2 and clathrin have distinct roles in cargo subcellular localization in *A. nidulans.*

## AP-2 co-localizes with DnfA, DnfB, AbpA, SagA and SlaB, but not with clathrin or UapA

We performed a series of co-localization studies examining the subcellular positioning of AP-2 in relationship to several membrane cargoes, including endocytic proteins (SagA and SlaB), actin polymerization markers (AbpA), apical markers (DnfA, DnfB and SynA) or transporters (UapA). For that we constructed strains expressing functional Ap2$^\sigma$-GFP or Ap2$^\sigma$-mRFP, and crossed these with strains expressing GFP- or RFP-tagged apical markers. *Figure 6A* shows that, in a wild-type background, Ap2$^\sigma$-GFP or Ap2$^\sigma$-mRFP has a polar cortical distribution, marking mostly the endocytic collar of growing tips, but also the septa. The apical and cortical localization of Ap2$^\sigma$-GFP was 'replaced' by a rather diffuse cytoplasmic fluorescent signal in an *ap2$^\mu$Δ* genetic background, showing that disruption of the full AP-2 complex by deleting a single subunit also disrupts its physiological localization. Noticeably, apical localization of Ap2$^\sigma$-GFP remained unaffected in a genetic background lacking clathrin light chain (repressed *thiA$_p$-claL*). Kymograph analysis further showed that the AP-2 complex marking the apical region, including the collar, is rather static. This also suggests that AP-2 is not localized in highly motile early endosomes (EEs). Additionally, no significant co-localization of Ap2$^\sigma$-GFP was obtained with a marker of the TGN (PH$^{OSBP}$) (*Figure 6—figure supplement 1*).

AP-2 co-localized cortically, with a high statistical significance (p<0.001), with SagA (80.7%), SlaB (77.4%), DnfA (70.1%), DnfB (77.0%), AbpA (75.8%) and SynA (68.8%). We also quantified and statistically confirmed that co-localization of AP-2 with apical markers (DnfA and SynA) occurs basically in the collar region, rather than the apical tip (lower panel in *Figure 6C*). Noticeably, co-localization of AP-2/SlaB or AP-2/SagA was modified in genetically deleted *sagAΔ* or *thiA$_p$-slaB* backgrounds, respectively, becoming less polarized and more extending away from the tip (*Figure 6D*). Importantly, AP-2 did not co-localize with clathrin (ClaL) or UapA (*Figure 6B,C*). In fact ClaL and UapA, in contrast to all other apical markers, do not localize significantly in the collar region, where apical endocytosis occurs.

Given the importance of confirming the non-colocalization of AP-2 and clathrin for the present work, we obtained additional evidence to support this finding. *Figure 6—figure supplement 2*, panel A, shows that strains expressing the chimeric ClaL-mRFP or ClaL-GFP proteins grow similar to wild-type, contrasting the severe growth defects associated with the lack of ClaL expression. This

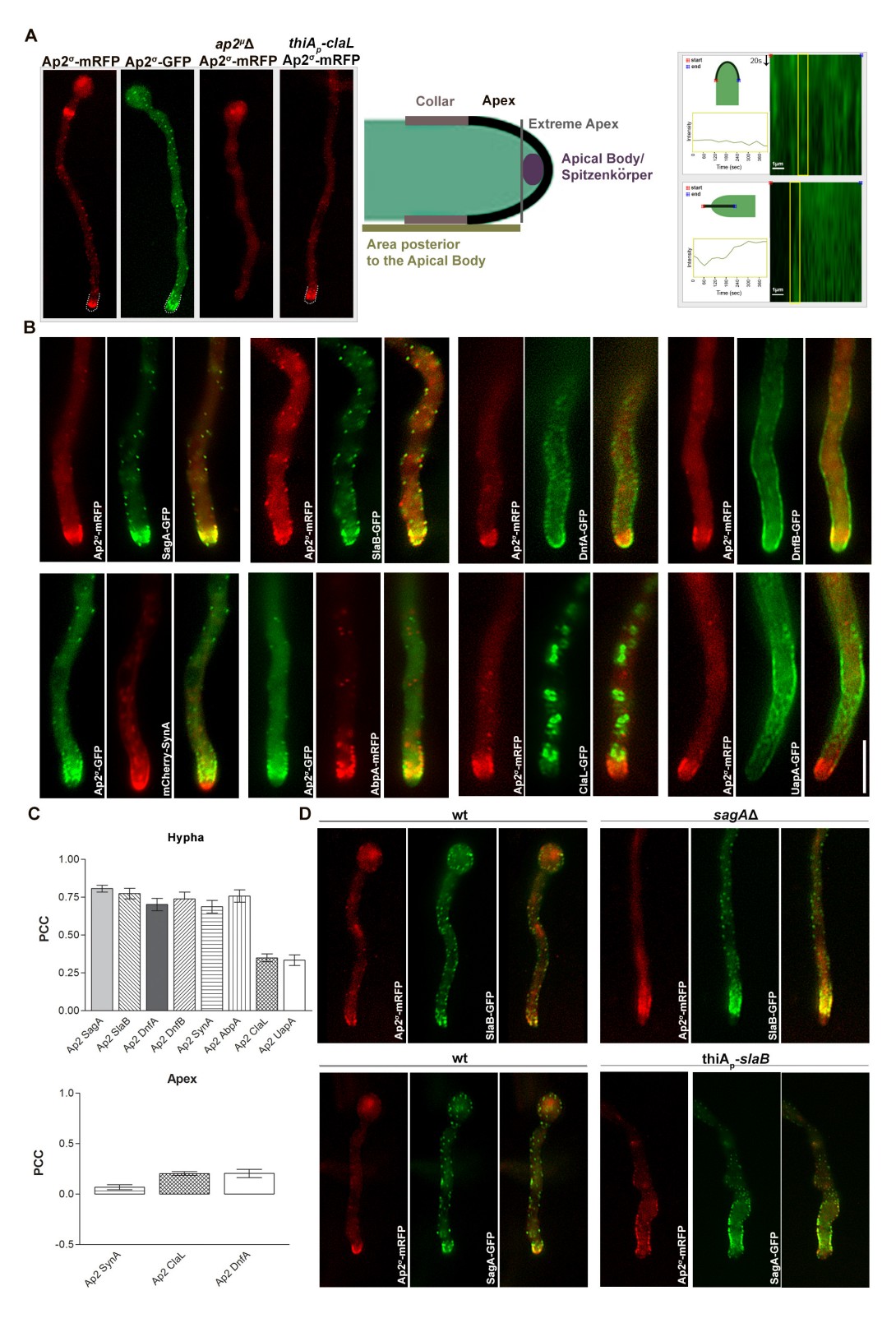

**Figure 6.** AP-2 shows polar co-localization with DnfA, DnfB, SagA and SlaB, but not with clathrin or UapA. (**A**) Subcellular localization of functional Ap2σ-mRFP or Ap2σ-GFP in wild-type background, or of Ap2σ-mRFP in *ap2ᵘΔ* or *thiAₚ-claL* backgrounds. Notice that the absence of a functional μ subunit leads to non-polar and non-cortical fluorescent signal of Ap2σ-mRFP, whereas Ap2σ-mRFP remains apically localized in the absence of clathrin (left panel). Biological/Technical replicates: 5/10, 5/10, 3/6 and 3/12, respectively. Cartoon depicting the hyphal tip of *A. nidulans* (middle panel).
*Figure 6 continued on next page*

*Figure 6 continued*

Kymograph analysis showing the rather static localization of Ap2$^\sigma$-GFP at the hyphal tip (right panel). Biological/Technical replicates: 2/3. (**B**) Subcellular localization experiments related the possible co-localization of Ap2$^\sigma$-mRFP or Ap2$^\sigma$-GFP with GFP- or mRFP-tagged SagA, SlaB, DnfA, DnfB, SynA, AbpA, ClaL and UapA. Notice the apparent cortical co-localization, especially at the collar region, of AP-2 with SagA, SlaB, DnfA, DnfB, SynA and AbpA, but not with ClaL or UapA. Biological replicates: 2, Technical replicates: 5–7. (**C**) Quantification of co-localization by calculating Pearson's Correlation Coefficient (PCC) for n = 5 hyphae, confirming significant co-localization of AP-2 with SagA, SlaB, DnfA, DnfB, SynA and AbpA. P-values are p<0.0001 for co-localization of AP-2 with SagA, SlaB, DnfA, DnfB, SynA, AbpA, p=0.0002 and p=0.0007 for ClaL and UapA respectively (upper panel). Quantification of co-localization by calculating Pearson's Correlation Coefficient (PCC) specifically at the apical region of tips for n = 7, 10, 6 tip regions of strains co-expressing fluorescent-tagged AP-2 and DnfA, SynA, ClaL respectively, showing that AP-2 does not co-localize with SynA, ClaL or DnfA (lower panel). P-values are 0.0026, 0.0250 and 0.0001 respectively. See Materials and methods for statistical analysis methods and statistical tests used. (**D**) Subcellular co-localization of Ap2$^\sigma$-mRFP with SlaB-GFP or SagA-GFP in *sagAΔ* or *thiA$_p$-slaB* backgrounds, respectively. Notice the relative depolarization of Ap2$^\sigma$-mRFP/SlaB-GFP in *sagAΔ* and of Ap2$^\sigma$-mRFP/SagA-GFP in *thiA$_p$-slaB*. Representative phenotypes selected from 20 hyphae for wt and mutant strains. Biological replicates: 2, Technical replicates: 10.

The following figure supplements are available for figure 6:

**Figure supplement 1.** Epifluorescence microscopy following the in parallel localization of Ap2$^\sigma$-GFP and the TGN marker mRFP-PH$^{OSBP}$.

**Figure supplement 2.** Evidence for the functionality and proper subcellular localization of GFP- or mRFP-tagged ClaL.

constitutes good evidence that GFP- or RFP-tagged versions of ClaL are functional. Additionally, panel B shows that the localization of ClaL-GFP or ClaL-mRFP is identical and compatible with the expected localization for clathrin, as in both cases ClaL marks Golgi-like structures and cortical foci. Panel C further shows a rather weak association of ClaL with subcortical patches close to the collar endocytic region. The apparent prominent localization of ClaL in Golgi-like structures is in full agreement with results in *Schultzhaus et al. (2017b)*, who have shown that ClaH-GFP localizes principally in the late Golgi, and has only a weak association with the sub-apical collar region. In the same article, the authors have further shown that ClaH and ClaL co-localize. Finally, the functionality of ClaL-mRFP was also confirmed by showing that strains expressing ClaL-mRFP are fully active in respect to UapA-GFP endocytosis (panel D), contrasting the block of UapA endocytosis when ClaL is not functional (see *Figure 4G*).

To further test the distribution of AP-2 and clathrin between plasma membrane and internal vesicles, as well as, to assess more rigorously the degree of possible co-localization between these two protein complexes, we analysed the localization of Ap2$^\sigma$-GFP and ClaL-mRFP by TIRF microscopy, followed by relevant quantification analysis (*Figure 7*). Practically no co-localization of the two polypeptides was observed (PPC = 0.12) in the PM. Only a minor fraction of the two polypeptides colocalized (PPC = 0.34) intracellularly. These data are in perfect line with the rest of our findings, confirming that the function of AP-2 in apical endocytosis is clathrin-independent.

Overall, our results are also in line with the following notions. First, AP-2 seems to synergize with endocytic factors SlaB and SagA at the endocytic collar, at sites of actin polymerization, marked by AbpA. Second, disruption of SagA or SlaB, but not of AP-2, somehow depolarizes the localization of this endocytic complex. Third, AP-2 is very probably the cargo-recognition (for example, DnfA and DnfB) partner of this complex. Finally, AP-2 and clathrin are involved in mutually exclusive endocytic or secretion pathways, also supporting a cargo-centric view of membrane trafficking (*Maldonado-Báez et al., 2013*).

## Further genetic and cellular evidence that AP-2 is involved in apical lipid maintenance

To further establish the role of AP-2 we tested its genetic interactions with SagA, DnfA and DnfB, but also with proteins involved in apical lipid maintenance, namely StoA and BasA. StoA is a stomatin homologue involved in the maintenance of apical sterol-rich membrane domains (SRDs) and polarity in *A. nidulans* (*Takeshita et al., 2012*). In metazoa, stomatins are oligomeric, lipid raft-associated proteins with scaffolding functions necessary for the maintenance of specific lipid composition in membranes, but their molecular function is still unclear (*Lapatsina et al., 2012*). BasA is required for phytosphingosine biosynthesis and is essential for fungal viability. A previously reported

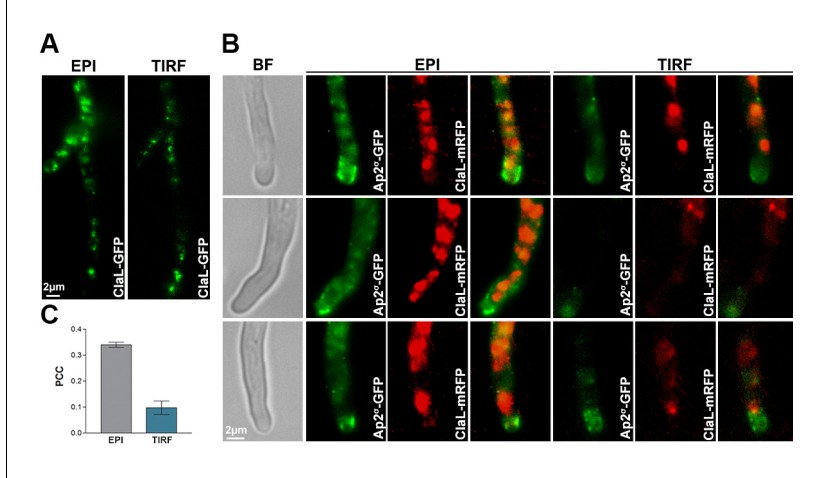

**Figure 7.** TIRF Microscopy confirms the non-colocalization of clathrin and AP-2. (**A**) Epifluorescence images and respective TIRF (Total Internal Reflection Fluorescence) microscopy of ClaL-GFP, confirming that a fraction of ClaL is associated with the PM. The penetration depth for TIRF was set to 110 nm. Biological/Technical replicates: 2/10. (**B**) Additional subcellular localization experiments investigated the possible co-localization of Ap2$^\sigma$-GFP with ClaL-mRFP. Epifluorescence microscopy (EPI) confirms the very low cortical co-localization of AP-2 with ClaL, similar to that observed when the two proteins where inversely tagged (see **Figure 6C**). Respective TIRF microscopy shows no co-localization of the two proteins in the plasma membrane. The penetration depth for TIRF was set to 150 nm. Biological replicates: 3, Technical replicates: 8. BF: Brightfield. (**C**) Quantification of co-localization by calculating Pearson's Correlation Coefficient (PCC) for n = 3 hyphae, of a strain co-expressing GFP-tagged AP-2 and ClaL-mRFP. The corresponding P-values are p<0.0001 and p<0.05 for the PCCs calculated by epifluorescence microscopy and by TIRF, respectively. See Materials and methods for statistical analysis methods and statistical tests used.

thermosensitive missense mutation in *basA* (*basA1*) resulted in an aberrant cell wall thickening and growth arrest at 42°C (*Li et al., 2007*). For the present work, we used null mutants of SagA (*Karachaliou et al., 2013*), DnfA, DnfB (*Schultzhaus et al., 2015*) and StoA (*Takeshita et al., 2012*), and either *basA1* or a conditional knock-down mutant of *basA*, constructed herein using the *thiA$_p$* promoter. These mutants were all crossed with the *ap2$^\sigma$Δ* strain. Growth phenotypes of the resulting double mutants compared to single mutants are shown in **Figure 8A and D**. In all cases, double mutants showed reduced growth compared to single mutants (that is, synthetic negative phenotypes), strongly suggesting that the function of AP-2 is related to that of SagA, DnfA and DnfB, as expected based on the subcellular localization experiments, but also to StoA and BasA. Microscopic examination of the morphology of the mutants confirmed that the defects in AP-2 and SagA, DnfA, DnfB, StoA or BasA were additive (**Figure 8B,C and E**). Finally, **Figure 8F** shows that the localization of AP-2$^\sigma$-GFP in a *basA1* genetic background is not anymore in the collar region, but instead marks internal structures just behind the tip. All above observations were confirmed by relevant quantitative analysis, as shown in **Figure 8G and H**.

Given that AP-2 was related to BasA, and thus sphingolipid localization, we tested whether AP-2 also affects the localization of ergosterol, the partner of sphingolipids in lipid rafts. To do so, we used filipin III, an established fluorescent ergosterol marker (*Van Leeuwen et al., 2008*). In wild-type *A. nidulans* and other filamentous fungi, filipin stains the PM, but with a predominant polar deposition at the hyphal apex (*Li et al., 2006*). We observed significant depolarization, often associated with the appearance of discrete cortical foci and abnormal filipin staining in the *ap2$^\sigma$Δ*, as well as, in *thiA$_p$-slaB*, *stoAΔ* and *thiA$_p$-basA* mutant backgrounds. In contrast, we observed a polar localization of filipin in *thiA$_p$-claL*, *dnfAΔ*, *dnfBΔ* and *sagAΔ* mutant strains, similar to wild-type (**Figure 9A**). Quantification of these results, by measuring the strength of the filipin fluorescent signal along the tip, confirmed the critical role of AP-2, as well as, of BasA, SlaB or StoA, in the apical depositioning of ergosterol (**Figure 9B**). These results further supported that AP-2 has a specific role in lipid apical localization, distinct from clathrin, necessary polar growth.

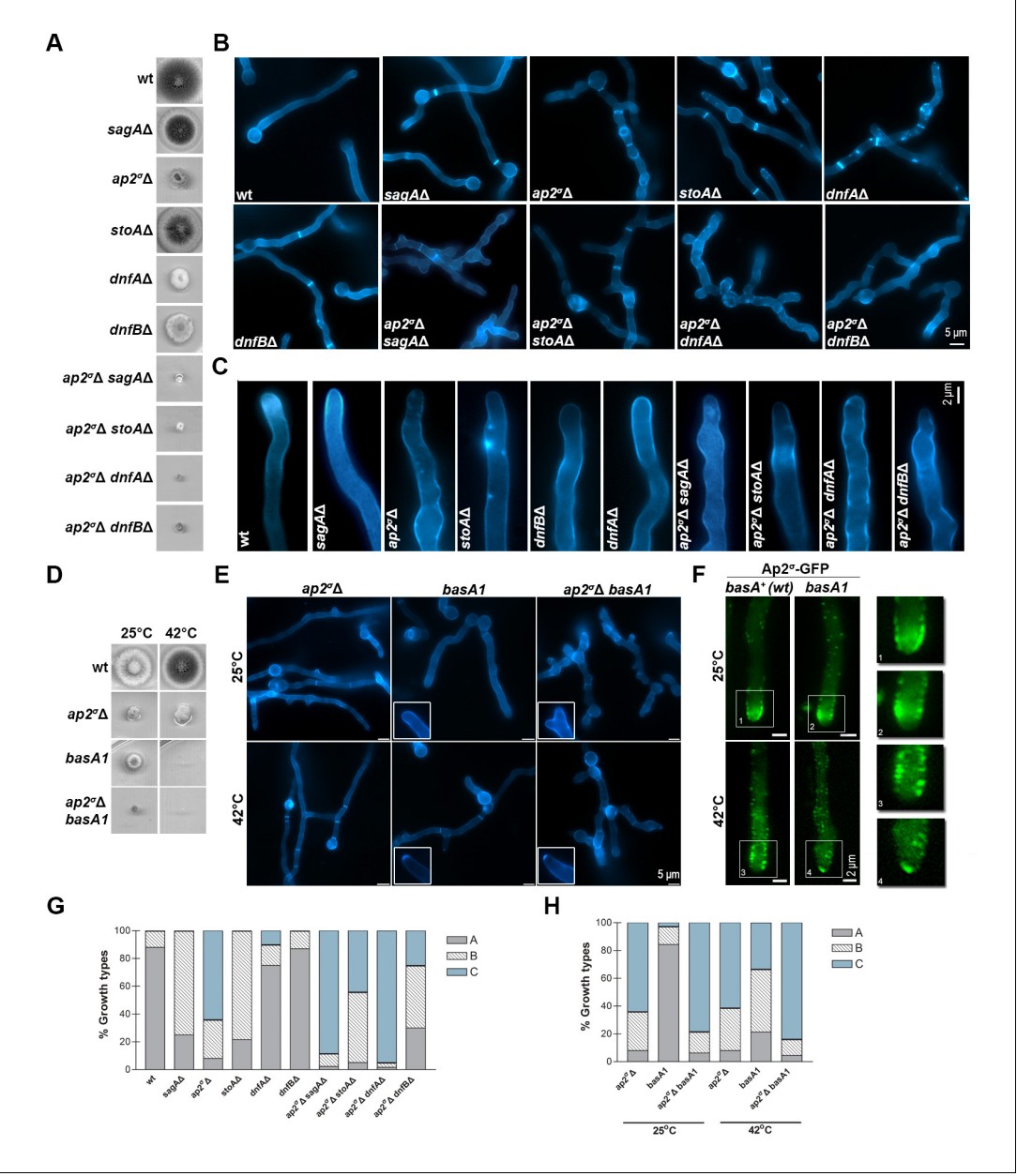

**Figure 8.** AP-2 interacts genetically with endocytic factors and proteins involved in apical lipid maintenance. (**A**) Growth phenotypes of single and double null mutants related to AP-2 and SagA, StoA, DnfA and DnfB. (**B**) Microscopic morphology of hyphal cells, stained with calcofluor, of strains shown in (**A**). Representative phenotypes selected from 100 hyphae for wt and 20–50 hyphae for mutant strains. Biological/Technical replicates: 4/25, 3/10, 3/15, 2/20, 2/15, 2/15, 3/10, 2/20, 2/15 and 2/15, respectively. (**C**) Apical deposition of calcofluor in strains shown in (**A**) and (**B**). Biological/Technical replicates as in (**B**). (**D, E**) Growth phenotypes and microscopic morphology of $ap2^{\sigma}\Delta$, $basA1^{ts}$ and $ap2^{\sigma}\Delta$ $basA1^{ts}$ strains. Inserts highlight the modification of calcofluor deposition from the collar region to the extreme apex in $basA1^{ts}$ strains under the non-permissive temperature (42°C). Representative phenotypes selected from 45 hyphae for $ap2^{\sigma}\Delta$ and mutant strains. Biological/Technical replicates: 3/15. (**F**) Localization at the extreme apex, rather than in the collar region, of AP-2 in a $basA1^{ts}$ genetic background. Notice that a similarly modified localization of calcofluor (chitin) was obtained in the $basA1^{ts}$ at 42°C (see relevant inserts in *Figure 7E*). Representative phenotypes selected from 30 hyphae. Biological replicates: 2, Technical replicates:15. (**G–H**) Quantitative analysis of growth types shown in (**B**) and (**E**), categorized as in *Figure 2* in A, B or C. (**G**) Analysis of n = 100 hyphae of wild-type and n = 32, 58, 51, 40, 92, 43, 59, 58, 100 hyphae of mutant strains. (**H**) Analysis of n = 25, 77, 65 and n = 75, 42, 68 hyphae of $ap2^{\sigma}\Delta$, $basA1$, $ap2^{\sigma}\Delta$ $basA1$ at 25°C and 37°C respectively.

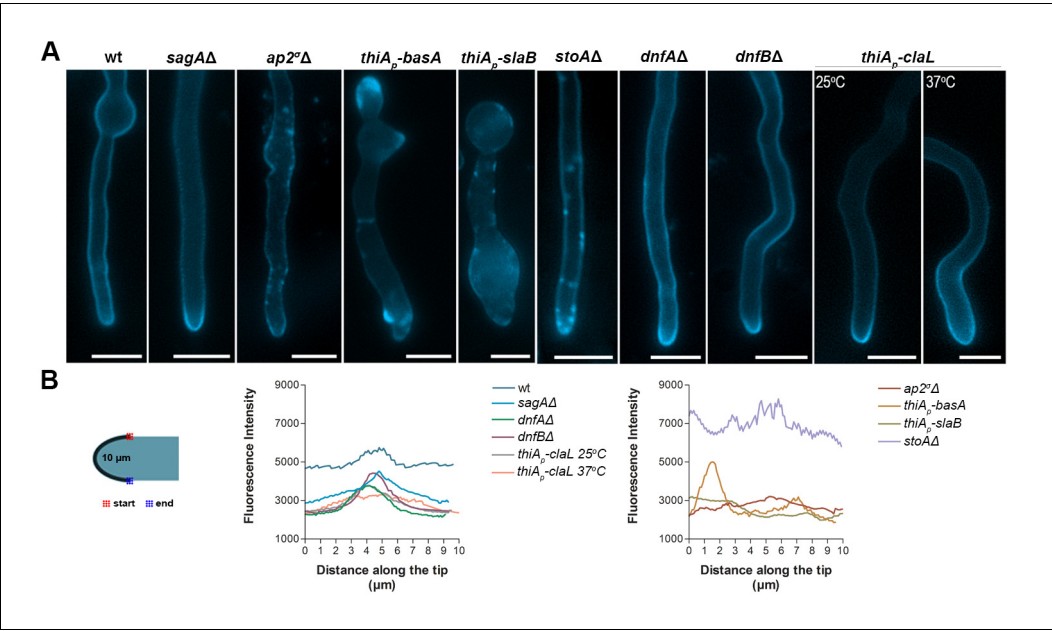

**Figure 9.** AP-2 is critical for ergosterol membrane localization. (**A**) Apical filipin staining of ergosterol in hyphal cells of wild-type and mutants. Notice the significant alterations and loss or reduction in apical staining in *ap2ᵒΔ*, *thiA_p-basA, thiA_p-slaB* and *stoAΔ* genetic backgrounds. Representative phenotypes selected from n = 20 hyphae for wt and mutant strains. Biological replicates: 2, Technical replicates: 20. Scale bars represent 5 µm. (**B**) Quantitative analysis of fluorescence intensity of filipin staining along the surface of hyphae tips. The region measured is depicted in the cartoon on the left.

## Discussion

In mammals, the AP-2 appendages of α2 and β2 subunits orchestrate endocytosis through hierarchical interactions with clathrin, other endocytic proteins and PM lipids. The appendage of the β2 subunit specifically interacts with clathrin (*Keyel et al., 2008*; *Thieman et al., 2009*). Lower eukaryotes, like free living filamentous fungi, face the challenge of rapid and polarized growth via apical extension, a process that is absolutely dependent on efficient endocytosis and recycling of chaperones and enzymes, related to PM and cell wall deposition at the growing tip (*Peñalva, 2010*, *2015*; *Peñalva et al., 2012*). Fungi also need both basal and conditionally elicited endocytosis all along their hyphal PM, serving the renewal or modification of membrane components in response to changing nutritional or stress conditions, best exemplified by the endocytic turnover of transporters (*Gournas et al., 2010*; *Karachaliou et al., 2013*; *Diallinas, 2014*). The role of AP-2 and that of clathrin in the endocytosis of apical cargoes or transporters had not been studied in filamentous fungi, until recently.

Here we showed that AP-2 is dispensable for transporter endocytosis and conventional apical secretion, but essential for polarity maintenance and the polar localization of membrane lipid or cell wall components. In parallel, we showed that clathrin (both ClaH and ClaL) is also essential for *A. nidulans* growth and that ClaH has a critical role in Golgi function and cargo secretion, which is probably the main reason why the ClaH null mutation is lethal. The importance of ClaH in conventional secretion via the Golgi was also recently reported by *Schultzhaus et al. (2017b)*. An essential role of clathrin in the endocytosis of transporters is also well supported by our results. Importantly, we also showed that the function of ClaL is unrelated to polarity maintenance or the polar localization of lipid or cell wall- related apical cargoes, which contrasts its essential role in transporter endocytosis. Our results also supported that the effect of ClaH depletion in maintaining the polar localization of apical cargoes is due to malfunctioning of Golgi-dependent secretion, rather than a direct effect on the polar localization establishment, similarly to what is observed when BFA is added to growing cells. These results, together with the absence of any significant overlap in the PM localization of

clathrin and AP-2, and the distinct phenotypes of clathrin and AP-2 mutants, confirmed that clathrin and AP-2 are recruited in distinct, cargo-dependent, trafficking pathways.

Led by the observation that genetic knock-out of AP-2 resulted in phenotypes compatible with loss of polarity maintenance, we followed the effect of deleting AP-2 on the subcellular localization of well-established apical markers. We thus showed that AP-2 co-localizes apically with endocytic factors SlaB and SagA at sites of actin polymerization in the sub-apical collar region. Subsequently, we provided direct evidence that AP-2 has a role on the apical localization and function of DnfA and DnfB flippases, and affects ergosterol (filipin) and cell wall (calcofluor) deposition at the tip. We finally showed that AP-2 interacts functionally with other proteins involved in apical lipid mainte-nance and scaffolding (BasA and StoA). The key role of AP-2 in apical endocytosis is in line with pre-vious results showing that proper membrane lipid and cell wall composition is essential for fungal polar growth (*Cheng et al., 2001*; *Takeshita et al., 2012*). A speculative model on the role of AP-2 and other relevant factors in apical endocytosis and polarity maintenance is shown in *Figure 10*. This model also considers recent findings on the recycling of apical markers described in *Schultzhaus et al. (2015)*; *(2016)*; *(2017b)* and *Peñalva (2015)*.

An AP-2 independent role of clathrin in endocytosis is not novel, as it has been reported before in mammals and yeast (*Conner and Schmid, 2003*; *Motley et al., 2003*; *Sorkin, 2004*; *Traub, 2009*; *Brach et al., 2014*). To our knowledge, however, the opposite is a novel finding, as no other report has shown a major clathrin-independent role of the AP-2 complex. A role, albeit minor, for AP-2 in maintaining normal post-endocytic trafficking of major histocompatibility complex class I (MHCI) pro-teins and beta1 integrin is the only reported case of a clathrin-independent role of AP-2 (*Lau and Chou, 2008*). Interestingly, Microsporidia possess AP complexes, but not clathrin, which further shows that AP complexes can function without clathrin. The fact that no canonical clathrin binding domains were identified in the $\beta$1 and $\beta$2 subunits of primitive fungi supports the notion that Dikarya have lost clathrin-binding domains during their evolution. The experimental support of a clathrin-independent role of AP-2 in endocytosis was in good agreement with the in silico observation that the $\beta$2 subunit of AP-2 of all higher fungi lacks the entire C-terminal $\beta$ appendage, which includes known clathrin binding domains. Interestingly, clathrin binding domains are also missing from the AP-1 $\beta$ subunit ($\beta$1) of all Dikarya, suggesting that in fungi AP-1 might also function independently of clathrin, very probably being involved in the recycling via the TGN compartment of specific apical proteins. Thus AP-2 and AP-1 might function in the same pathway for apical cargo recycling. Inter-estingly, AP-1 has been reported to function independently of clathrin in phagocytosis in murine macrophages (*Braun et al., 2007*). This has been correlated with the presence of two isoforms of the $\gamma$-adaptin subunit of AP-1 in macrophages (*Santambrogio et al., 2005*), and it has been pro-posed that the cleaved form of AP-1 is the one associated with vesicles present under phagocytic cups (*Braun et al., 2007*).

Our findings strongly suggest the existence of clathrin-independent endocytosis (CIE) related to polar growth in *A. nidulans*. Over the last years, there has been an increasing interest concerning CIE, not only because it is the mode of entry of bacterial toxins and cell surface proteins (*Maldo-nado-Báez et al., 2013*; *Mayor et al., 2014*), but also because it has raised strong debates concern-ing its physiological significance. In a recent report, *Bitsikas et al. (2014)*, provided evidence that clathrin-independent pathways do not, in fact, contribute significantly to endocytic flux. However, in cases where it has been supported experimentally, CIE seems to be specialized for the maintenance of PM lipid composition (*Shvets et al., 2015*) and the endocytosis of proteins anchored to the mem-brane by glycosyl phosphatidylinositol (GPI) (*Nichols, 2009*; *Bitsikas et al., 2014*). Additionally, in some cases CIE has also been reported to depend on specific PM lipid rafts and/or microdomains, such as caveolae or flotillins (*Glebov et al., 2006*), despite some recent contradicting reports sup-porting that caveolae and flotillins function in organizing PM domains, rather being *bona fide* endo-cytic factors (*Parton and del Pozo, 2013*). Finally, some CIE pathways also depend on dynamin (*Lamaze et al., 2001*). An apparent conclusion from the above and other studies is that distinctions and variations in both clathrin-dependent endocytosis (CME) or CIE seem to arise from the endocytic cargo being examined (*Maldonado-Báez et al., 2013*). Our work suggests that AP-2 is a key cargo-specific recognition factor in apical CIE in fungi.

For long, yeasts were thought to depend solely on CME. However, recent evidence demon-strated the existence of a CIE pathway that depends on the GTPase Rho1, Rom1/2, formin Bni1 and $\alpha$-arrestins (*Prosser and Wendland, 2012*; *Prosser et al., 2011*, *2015*). In addition, a CIE pathway

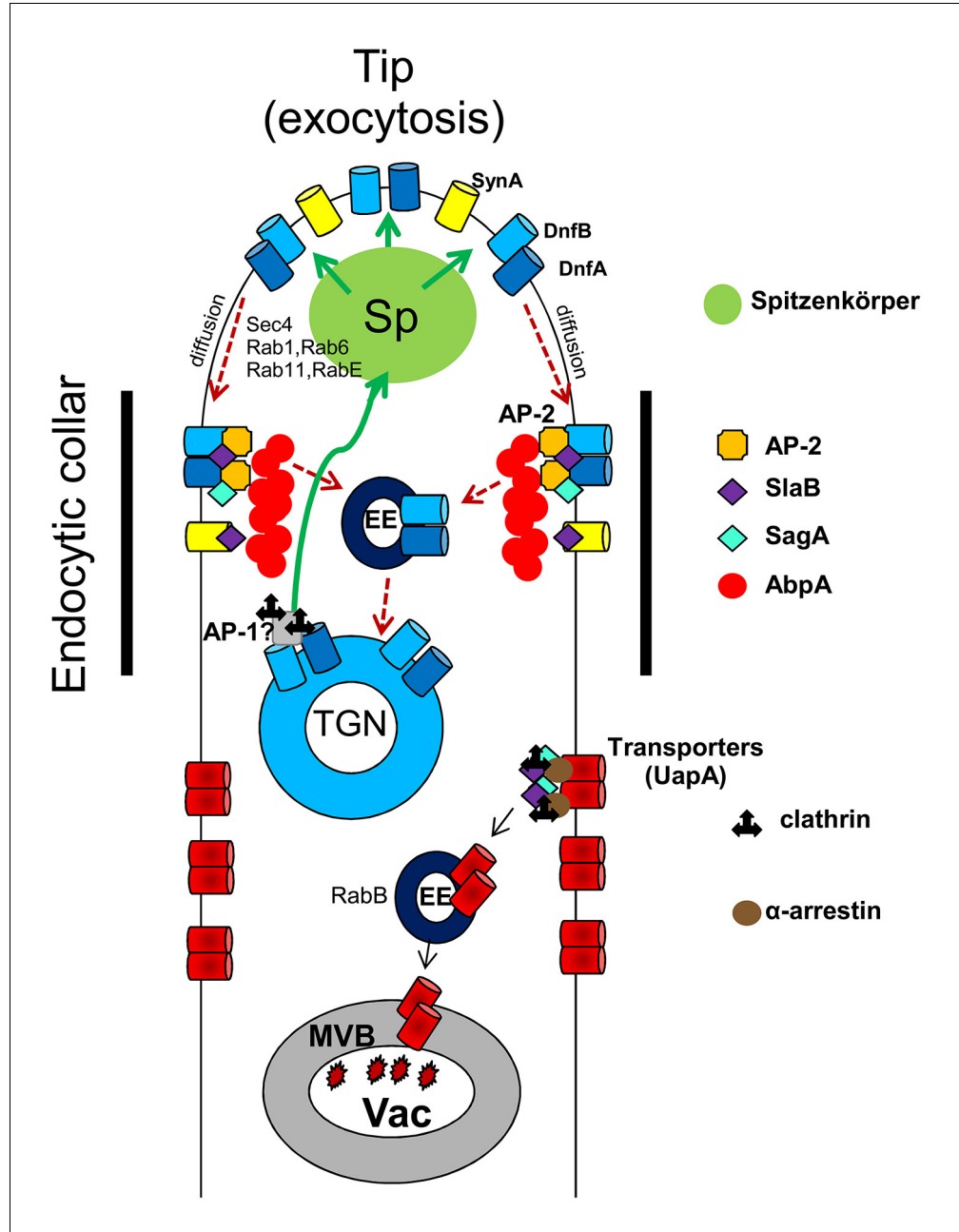

**Figure 10.** A speculative model highlighting the role of AP-2 in DnfA and DnfB endocytosis at the apical region of *A. nidulans* growing hyphal cells. After reaching the PM, DnfA and DnfB diffuse laterally to the collar region where they are recognized by AP-2 and undergo actin polymerization-dependent endocytosis with the help of SagA and SlaB. Endocytic vesicles are sorted in the Early Endosomes (EEs) and from there DnfA and DnfB undergo retrograde traffic to the TGN, the Spitzenkörper (a vesicle sorting region in filamentous ascomycetes; *Pantazopoulou et al., 2014*) and eventually reach the PM (*Schultzhaus et al., 2015*; *Peñalva, 2015*). Extrapolating from the observation that AP-1 loss-of-function mutants are severely defective in polarity maintenance and growth, similar to AP-2 mutants, we predict that AP-1 is involved in the retrograde exocytosis (green arrows) of DnfA, DnfB and other cargoes essential for lipid or cell wall (for example, chitin synthases) maintenance. The model does not exclude that a fraction of DnfA and DnfB and other cargoes endocytosed by the AP-2 pathway would undergo degradation after being sorted into degradative EEs that are destined to the vacuole. The model also depicts that transporters and possibly other non-polar membrane proteins are not cargoes of the AP-2 pathway, but instead undergo clathrin- and α-arrestin-dependent endocytosis, followed by sorting into degradative EEs and eventual degradation in the vacuole (Vac). The model also suggests that SlaB

*Figure 10 continued on next page*

*Figure 10 continued*

and SagA endocytic factors might have roles in both the AP-2 and clathrin endocytic pathways. Finally, the model shows that SynA (V-Snare) is not a cargo of the AP-2 pathway. The roles of some other factors in exocytosis (Sec4, Rab11, Rab1, Rab6 and RabE) or sorting (RabB) of cargoes, is based on the work of the group of M.A. Peñalva (*Peñalva, 2015*).

was discovered in *Candida albicans* (*Epp et al., 2013*). Interestingly, in yeast, α-arrestins are important for cargo selection in both the CME and CIE, but seem to function by distinct mechanisms in each pathway, as in the CIE pathway, unlike CME, their function is independent of the Rsp5 ubiquitin ligase (*Lin et al., 2008*; *Nikko et al., 2008*; *Prosser et al., 2015*). Although the involvement of AP-2 in CIE in yeast has not been examined, the involvement of α-arrestins differentiates it from the apical CIE pathway we identify herein, as none of the *A. nidulans* arrestin knock-out mutants shows defects in polarity maintenance and growth (*Karachaliou et al., 2013*). This however does not exclude the possibility that the apical CIE pathway in *A. nidulans* shares other factors of the CIE pathway identified in yeast (for example, Rho1, Rom1/2 or formin Bni1). In fact, SepA (orthologue of Bni1; *Harris et al., 1997*; *Sharpless and Harris, 2002*), is necessary for septum formation and polarity maintenance, and RhoA$^{Rho1}$ plays a role in polarity, proper branching pattern, and cell wall deposition (*Guest et al., 2004*), phenotypes related to the proposed AP-2 function.

The present work does not intend to explore the mechanistic details of the AP-2 pathway in *A. nidulans* or identify the entire set of factors involved. However it shows, for the first time, the existence of a clathrin-independent pathway related to polarity maintenance in filamentous fungi, and most importantly identifies AP-2 as a key factor for the apical recycling of enzymes, lipids and cell wall components necessary for fungal growth. From the evolutionary point of view, our work suggests that in higher fungi the function of the AP-2 complex has been uncoupled, probably via a specific gene truncation, from clathrin-mediated endocytosis, and that this uncoupling seems to serve specific cellular challenges of fungal polar growth.

## Materials and methods

### Media, strains, growth conditions and transformation

Standard complete and minimal media (MM) for *A. nidulans* were used. Media and supplemented auxotrophies were at the concentrations given in FGSC (http://www.fgsc.net.) (RRID: SCR_008143). Media and chemical reagents were obtained from Sigma-Aldrich (Life Science Chemilab SA, Hellas) or AppliChem (Bioline Scientific SA, Hellas). Glucose 1% (w/v) was used as a carbon source. NaNO$_3$ at 10 mM was used as a nitrogen source. Thiamine hydrochloride was used at a final concentration of 5–10 μM. *A. nidulans* transformation was performed as described previously in *Koukaki et al. (2003)*. An *nkuA* DNA helicase deficient strain (TNO2A7; *Szewczyk et al., 2007*; *Nayak et al., 2006*) was the recipient strain for generating 'in locus' integrations of tagged gene fusions, or gene deletions by the *A. fumigatus* markers orotidine-5'-phosphate-decarboxylase (AF*pyrG*, Afu2g0836) or GTP-cyclohydrolase II (AF*riboB*, Afu1g13300), resulting in complementation of auxotrophies for uracil/uridine (*pyrG89*) or riboflavin (*riboB2*) respectively. Transformants were verified by PCR and Southern analysis. Combinations of mutations were constructed by standard genetic crossing. *A. nidulans* strains used are listed in *Supplementary file 1*.

### Standard nucleic acid manipulations and plasmid constructions

Genomic DNA extraction from *A. nidulans* was as described in FGSC (http://www.fgsc.net) (RRID: SCR_008143). Plasmid preparation from *E. coli* strains and DNA bands gel extraction were done using the Nucleospin Plasmid kit and the Nucleospin Extract II kit (Macherey-Nagel, Lab Supplies Scientific SA, Hellas). DNA sequences were determined by VBC-Genomics (Vienna, Austria). Southern blot analysis using specific gene probes and upstream or downstream fragments in the case of verifying gene deletions, was performed as described in *Sambrook et al. (1989)*. [$^{32}$P]-dCTP labeled molecules of gene specific probes were prepared using a random hexanucleotide primer kit following the supplier's instructions (Takara Bio, Lab Supplies Scientific SA, Hellas) and purified on

MicroSpin S-200 HR columns (Roche Diagnostics, Hellas). Labeled [$^{32}$P]-dCTP (3000 Ci mmol$^{-1}$) was purchased from the Institute of Isotops Co. Ltd, Miklós, Hungary. Restriction enzymes were from Takara Bio (Lab Supplies Scientific SA, Hellas). Conventional PCR reactions, high fidelity amplifications and site-directed mutagenesis were performed with KAPA Taq DNA and Kapa HiFi polymerases respectively (Kapa Biosystems, Lab Supplies Scientific SA, Hellas). Gene deletions and 'in locus' integrations of tagged gene fusions were generated by one step ligations or sequential cloning of the relevant fragments in the plasmids pBluescript SKII, or pGEM-T using oligonucleotides carrying additional restriction sites. These plasmids were used as templates to amplify the relevant linear cassettes by PCR. For primers and information related to these constructs see *Supplementary file 2*.

## Total protein extraction and western blot analysis

Cultures for total protein extraction were grown in MM supplemented with NaNO$_3$ at 25° C. Thiamine hydrochloride was used at a final concentration of 5–10 µM. Total protein extraction was performed as previously described (*Galanopoulou et al., 2014*). Equal sample loading was estimated by Bradford assays. Total proteins (30–50 µg) were separated in polyacrylamide gels (8–10 % w/v) and electroblotted (Mini PROTEAN Tetra Cell, BIORAD) onto PVDF membranes (Macherey-Nagel, Lab Supplies Scientific SA, Hellas). Immunodetection was performed with a primary mouse anti-GFP monoclonal antibody (Roche Diagnostics), a mouse anti-actin monoclonal (C4) antibody (MP Biomedicals Europe) and a secondary goat anti-mouse IgG HRP-linked antibody (Cell Signaling Technology Inc, Bioline Scientific SA, Hellas). Blots were developed using the LumiSensor Chemiluminescent HRP Substrate kit (Genscript USA, Lab Supplies Scientific SA, Hellas) and SuperRX Fuji medical X-Ray films (FujiFILM Europe). Quantification of ClaH-GFP or actin levels using ImageJ were estimated separately and relative to each other and are given as ratios, where in each case the lowest value was arbitrarily set as 1.

## Phylogenetic analysis

BLASTp searches were performed on the NCBI database (RRID: SCR_004870) to identify which organisms acquire a C-terminal domain on $\beta$ AP-2 subunit, with *H. sapiens* $\beta$ AP-2 as query. The selected sequences were retrieved and, with the use of UniProt database (RRID: SCR_002380), the C-terminal domains were identified and selected from each sequence. The phylogenetic tree reconstruction of the C-terminal domains was performed on MEGA6 software (RRID: SCR_000667) (*Tamura et al., 2013*) with the maximum-likelihood method and bootstrap testing for 150 replications. The substitution model was WAG and the ML heuristic method selected was Nearest-Neighbor-Interchange (NNI). NCBI sequences used for figure illustrations; *H. sapiens* (NP_001273: $\beta$AP-2, NP_001118: $\beta$AP-1 and NP_004635:$\beta$ AP-3), *F. alba* (XP_009492690: $\beta$ AP-2, XP_009492179: $\beta$ AP-1 and XP_009495754: $\beta$ AP-3), *R. allomyces* (EPZ36209: $\beta$ AP-2, EPZ33551: $\beta$ AP-1 and EPZ35993: $\beta$ AP-3), *S. punctatus* (KNC98413: $\beta$ AP-2, KNC96576: $\beta$ AP-1 and KND02292: $\beta$ AP-3), *A. nidulans* (CBF70501: $\beta$ AP-2, CBF83537: $\beta$ AP-1 and CBF90059: $\beta$ AP-3), *S. cerevisiae* (NP_012538: $\beta$ AP-2, NP_012787: $\beta$ AP-1 and NP_011777: $\beta$ AP-3), and *S. pombe* (NP_596435: $\beta$ AP-2, NP_595274: $\beta$ AP-1 and NP_593796: $\beta$ AP-3). The illustrations were taken from HMMER (RRID: SCR_005305) homology (search mode) of EBI database (RRID: SCR_002872) where the sequences were subjected and subsequently manipulated.

## Inverted epifluorescence microscopy, TIRF-M live imaging and statistical analysis

Samples for wide-field epifluorescence microscopy and Total Internal Reflection Fluorescence Microscopy (TIRF-M) were prepared as previously described (*Evangelinos et al., 2016*). Germlings were incubated in sterile 35 mm µ-dishes, high glass bottom (*ibidi*, Germany) in liquid MM for 16–22 hr at 25° C. Filipin III and Calcofluor white were used for 5 min prior to observation at final concentrations of 1 µg ml$^{-1}$ and 0001% (w/v) respectively. FM4-64 and CMAC staining was according to *Peñalva (2015)* and *Evangelinos et al. (2016)*, respectively. Brefeldin A was used at a final concentration of 100 µg ml$^{-1}$. Images were obtained using a Zeiss Axio Observer Z1/Axio Cam HR R3 camera. Contrast adjustment, area selection and color combining were made using the Zen lite 2012 software. Live imaging of plasma membrane ClaL was accomplished by TIRF-M. Hyphae were analyzed using a Leica AM TIRF MC set up on a Leica DMI6000 B microscope and a Leica 100X HCX PL

APO 1.4 NA objective. Biological replicates correspond to different samples, while technical replicates correspond to different hyphae observed and/or photographed within each sample. For kymograph generation and all measurements of fluorescence intensity, the *Reslice* and *Plot profile* commands in ImageJ (RRID: SCR_003070) were used, respectively (https://imagej.nih.gov/ij/). For the statistical analysis in *Figure 2*, Tukey's Multiple Comparison test was performed (One-way ANOVA), using the Graphpad Prism software (RRID: SCR_002798). Confidence interval was set to 95%. For quantifying colocalization (*Dunn et al., 2011*), Pearson's correlation coefficient (PCC) above thresholds, for a selected Region of interest (ROI) was calculated, using the coloc2 plugin of Fiji (RRID: SCR_002285). Costes P-value was 1.00 for all the images tested (*Costes et al., 2004*). PSF was set to 1.2 and the number of iterations was 100. One sample t-test was performed to test the significance of differences in PCCs, using the Graphpad Prism software. Confidence interval was set to 95%. Colocalization results were also confirmed using the ICY (RRID: SCR_010587) colocalization studio plugin (pixel-based method) (http://icy.bioimageanalysis.org/) for ROIs that included either the foci and the tip area which were selected using the Spot Detector plugin, or the extreme apex area, selected using the Area Selection tool. The same tool was also used for the measurement of Vacuolar Surface (Total surface of vacuoles containing GFP/Hypha) and Vacuolar GFP Fluorescence (Total fluorescence intensity of vacuoles containing GFP/Hypha) in *Figure 3E–F*, while Tukey's Multiple Comparison Test (One-Way ANOVA) using Graphpad Prism was performed to test the statistical significance of the results. Scale bars were added using the FigureJ plugin of the ImageJ (RRID: SCR_003070) software (*Mutterer and Zinck, 2013*). Images were further processed and annotated in Adobe Photoshop CS4 Extended version 11.0.2 (RRID: SCR_002078).

## Acknowledgements

We thank Brian Shaw for the DnfA and DnfB strains and Norio Takeshita for the StoA strain. We also thank Spiros Efthimiopoulos for Brefeldin A. This work was supported by the *Fondation Santé*, to which we are grateful.

## Additional information

### Funding

| Funder | Author |
| --- | --- |
| Fondation Sante | George Diallinas |

The funders had no role in study design, data collection and interpretation, or the decision to submit the work for publication.

### Author contributions

OM, Data curation, Software, Formal analysis, Investigation, Methodology; SA, Data curation, Investigation, Methodology, Writing—original draft; AZ, Software, Investigation; SC, Resources, Methodology; GD, Conceptualization, Resources, Formal analysis, Supervision, Funding acquisition, Validation, Visualization, Writing—original draft, Project administration, Writing—review and editing

### Author ORCIDs

George Diallinas, http://orcid.org/0000-0002-3426-726X

## Additional files

### Supplementary files

• Supplementary file 1. Strains used in this study.

• Supplementary file 2. Oligonucleotides used in this study for cloning purposes.

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
