## [Decision Letter]

Thank you for submitting your article "The AP-2 complex has a specialized clathrin-independent role in apical endocytosis and polar growth in fungi" for consideration by *eLife*. Your article has been reviewed by three peer reviewers, and the evaluation has been overseen by Randy Schekman who served as the Reviewing and the Senior Editor. The reviewers have opted to remain anonymous.

The reviewers have discussed the reviews with one another and the Reviewing Editor has drafted this decision to help you prepare a revised submission.

Summary:

This manuscript reports a clathrin-independent function for the AP-2 complex in filamentous fungi. In a bioinformatics analysis, they show that the β subunit of AP-2 of *A. nidulans*, as well as of other fungi, does not contain the C-terminal domain with the clathrin-binding site. They then investigate the role of AP-2 in *A. nidulans* endocytosis, and find that it is critical for polarized growth, but not for endocytosis of various plasma membrane-resident transporters; it has been known before that clathrin-mediated endocytosis does not necessarily require AP-2. However, they then show that polar localization of particular cargo requires AP-2 but, surprisingly, not clathrin. Furthermore, they show that polarized AP-2 colocalizes with a number of endocytic factors and polarized cargo, but not with clathrin or non-polarized cargo. Thus, they suggest that AP-2 has a (endocytic) role in polarization of the hyphal tip in filamentous fungi, which is independent of clathrin. A clathrin-independent endocytic role for AP-2 is a novel finding of broad interest to cell biologists. Although other AP complexes have been attributed clathrin-independent roles, these are not well understood. The here presented finding can contribute to our understanding of how endocytic adaptors function and of their specificity in cargo selection. Furthermore, it is interesting with regard to the role of clathrin function, especially its dispensability for some endocytic pathways. The manuscript is well written, the data well presented and the experiments appear to be carefully done.

Essential revisions:

1) One of the major findings is presented in Figure 5; the loss of polar localization of the flippases in absence of AP-2, but not in absence of clathrin. Given the importance of these results, they require quantification, and perhaps a higher number of representative example images. Would it be possible to quantify the fluorescence at the tips vs. fluorescence along the hyphae for mutants and wild type cells, for example by measuring the fluorescence intensities along line profiles at defined distances from the tips?

Throughout the manuscript, the data often consists of single fluorescence images of cell(s) from an experiment. When at all possible, quantification of multiple cells in multiple images should be carried out to support interpretations based on the images presented, as was done in Figure 2. For example, in the analysis of transporter uptake in Figure 3, the authors conclude that *ap2*∆ does not inhibit internalization (subsection “AP complexes are dispensable for membrane traffic and endocytosis of transporters”, last paragraph), but there is little apparent difference between control and endocytic trigger images of AgtA-GFP or FurA-GFP, suggesting that little internalization occurred in response to the endocytic trigger in these strains. Quantification of internalization or the level of transporter remaining at the cell surface could address this issue. In Figure 7, the authors conclude that the effects of double mutants on morphology is additive (subsection “Further genetic and cellular evidence that AP-2 is involved in apical lipid maintenance”, first paragraph) but most of the double mutant images resemble the single *ap2*∆ mutant. Some measurement of morphological phenotype is needed to support the conclusion that mutant effects are additive. In addition, Figure 4 would benefit from quantification of morphology and types of septa to allow comparison to *ap2*∆ mutants in Figure 2; Figure 5 would benefit from quantitation of the ratio of hyphae in each strain with polarized vs non-polarized protein distribution; Figure 8 would benefit from quantification of polarized distribution of filipin staining.

2) The reviewers are particularly concerned about the functionality of the ClaL-GFP construct and the extent of the depletion of the clathrin heavy and light chains in the experiments with thiamine. The author should know that C-terminal tagging of the clathrin light chain in *S. cerevisiae* results in a partially miss-functional protein, so they should provide evidence that ClaL-GFP is a fully functional protein. There is the possibility that ClaL depletion does not have a strong effect on triskelia, so they should heavy chain depletion. However, if the heavy chain depletion also does not affect flippase polarization, the clear genetic distinction between the pathways would be sufficient for an initial report.

*S. cerevisiae*.3) Another major concern is that the evolutionary significance of the findings is somewhat uncertain. The authors propose that clathrin-independent function applies to AP-2 in all dikarya (e.g. Abstract), and suggest that the AP-1 complex in dikarya also functions in a clathrin-independent manner (Discussion, third paragraph). These proposals are based primarily on the phylogenetic analysis in Figure 1, which shows that higher fungal species encode β subunits that are shorter than mammalian β subunits, lacking the folded domains in mammalian β appendages that are involved in binding clathrin and clathrin-accessory proteins. However, there is direct evidence that the S. cerevisaie AP-1 complex binds clathrin through the C-terminal regions of the β and γ subunits and functions in clathrin-dependent transport, even though the β subunit also lacks the folded appendage domains of mammalian β subunits (e.g. Yeung and Payne. 2001. Traffic 2(8):565-76). By extension, other fungal AP complexes with β subunits lacking folded appendage domains could have the potential to interact with clathrin through clathrin binding sequences in the non-conserved β "hinge" region and α appendage domain. Furthermore, *S. cerevisiae* AP-2 localizes to endocytic sites that are associated with clathrin (Carrol et al. 2009. Dev Cell 17(4):552-60, 2011; Carroll et al., 2012. Mol Biol Cell 23(4):657-68), making it likely that AP-2 functions with clathrin even though it does not exhibit strong clathrin binding (Yeung et al. 1999. Mol Biol Cell 10(11):3643-59). For these reasons, the authors' interpretation of the phylogenetic analysis in Figure 1 is questionable and it is difficult to predict whether clathrin-independent function described for *A. nidulans* will be a general feature of AP-2 in other higher fungal species. At a minimum, the manuscript should be rewritten to consider the data from *S. cerevisiae* and moderate any conclusions about other fungi.

4) A key indication for a spatial uncoupling of clathrin- and AP-2-dependent endocytosis is the colocalization data shown in Figure 6. ClCl^-^GFP (clathrin) does not appear to localize at the plasma membrane. What is seen in the micrograph is probably clathrin on intracellular membranes, which is likely to obscure weaker possible patches on the plasma membrane. However, to support the conclusions made, it is crucial to see where on the plasma membrane clathrin performs its endocytic function. Given that it is involved in transporter uptake like SlaB, one would expect, besides the expected intracellular localization, similar PM-patches for ClcL as for SlaB. Could the authors try to overcome this problem by using different imaging techniques, for example by TIRF microscopy, to visualize the plasma membrane only rather than the intracellular membranes? If not, it may be sufficient to assess colocalization of AP-2 and clathrin at endocytic sites by analyzing higher magnification/resolution images of the apex and collar regions.

The full comments of the reviewers follow. However, only the essential revisions listed above will be required for your revised submission. Additional major questions below may be addressed at your discretion.

*Reviewer #1:*

This manuscript reports a clathrin-independent function for the AP-2 complex in filamentous fungi. In a bioinformatics analysis, they show that the β subunit of AP-2 of *A. nidulans*, as well as of other fungi, does not contain the C-terminal domain with the clathrin-binding site. They then investigate the role of AP-2 in *A. nidulans* endocytosis, and find that it is critical for polarized growth, but not for bulk endocytosis of various plasma membrane-resident transporters. On the other hand, endocytosis of the same transporters appears to be clathrin dependent, which is also important for polarized growth. So far this is not surprising; it has been known before that clathrin-mediated endocytosis does not necessarily require AP-2. However, they then show that polar localization of particular cargo requires AP-2 but not clathrin. Furthermore, the show that polarized AP-2 colocalizes with a number of endocytic factors and polarized cargo, but not with clathrin or non-polarized cargo. Thus, they suggest that AP-2 has a (endocytic) role in polarization of the hyphal tip in filamentous fungi, which is independent of clathrin. A clathrin-independent endocytic role for AP-2 is a novel finding of broad interest to cell biologists. Although other AP complexes have been attributed clathrin-independent roles, these are not well understood. The here presented finding can contribute to our understanding of how endocytic adaptors function and of their specificity in cargo selection. Furthermore, it is interesting with regard to the role of clathrin function, especially its dispensability for some endocytic pathways. The manuscript is well written, the data well presented and the experiments appear to be carefully done. The following points should be addressed:

1) One of the major findings is presented in Figure 5; the loss of polar localization of the flippases in absence of AP-2, but not in absence of clathrin. Given the importance of these results, they require quantification, and perhaps a higher number of representative example images. Would it be possible to quantify the fluorescence at the tips vs. fluorescence along the hyphae for mutants and wild type cells, for example by measuring the fluorescence intensities along line profiles at defined distances from the tips?

2) The involvement of well-characterized endocytic markers in the two proposed pathways should be established more thoroughly, as it is important to know if the two pathways share part of their machinery. The best-characterized fungal endocytic machinery is the one of budding yeast *S. cerevisiae*. Firstly, does SlaB, which the authors use as endocytic marker, correspond to Sla2 in *S. cerevisiae*? What is the *S. cerevisiae* ortholog of SagA?

If SlaB corresponds to Sla2, this would a good marker for endocytosis, especially as it is known to be essential for endocytosis in budding yeast (Kaksonen et al., Cell 2003). Are SlaB and SagA involved in both, the AP-2- and clathrin-dependent pathways in *A. nidulans*? The colocalization studies may suggest this, but are not very strong evidence. The authors show that UapA internalization depends on SlaB and SagA, but not whether polar localization of the flippases DnfA and DnfB does. They could use the SlaB and SagA deletion/knockdown mutants to test this.

Along the same lines, it is a bit confusing that FM4-64 uptake is neither affected by lack of AP-2, clathrin, SagA nor SlaB. This would actually suggest that SagA and SlaB are not involved in both pathways, or that there is a third pathway for uptake. Can the authors experimentally test these possibilities?

3) A key indication for a spatial uncoupling of clathrin- and AP-2-dependent endocytosis is the colocalization data shown in Figure 6. ClCl^-^GFP (clathrin) does not appear to localize at the plasma membrane. What is seen in the micrograph is probably clathrin on intracellular membranes, which is likely to obscure weaker possible patches on the plasma membrane. However, to support the conclusions made, it is crucial to see where on the plasma membrane clathrin performs its endocytic function. Given that it is involved in transporter uptake like SlaB, one would expect, besides the expected intracellular localization, similar PM-patches for ClcL as for SlaB. Could the authors try to overcome this problem by using different imaging techniques, for example by TIRF microscopy, to visualize the plasma membrane only rather than the intracellular membranes?

4) The Calcofluor staining in Figure 2 is important, as it shows depolarization of apical endocytosis in the *ap2* mutant. However, the images are not entirely convincing, and the conclusions would be much better supported by a quantification similar to that suggested in point 1 (for Figure 5).

*Reviewer #2:*

This manuscript characterizes the role of the *Aspergillus nidulans* AP-2 adaptor complex in endocytosis using genetic and localization approaches. The results provide evidence that, in this filamentous fungus, AP-2 functions in cargo-selective apical endocytosis independently of clathrin. This is a novel observation, which suggests for the first time (to my knowledge) that AP-2 has evolved in an organism to function independently of clathrin.

One major concern is that the evolutionary significance of the findings is somewhat uncertain. The authors propose that clathrin-independent function applies to AP-2 in all dikarya (e.g. Abstract), and suggest that the AP-1 complex in dikarya also functions in a clathrin-independent manner (Discussion, third paragraph). These proposals are based primarily on the phylogenetic analysis in Figure 1, which shows that higher fungal species encode β subunits that are shorter than mammalian β subunits, lacking the folded domains in mammalian β appendages that are involved in binding clathrin and clathrin-accessory proteins. However, there is direct evidence that the S. cerevisaie AP-1 complex binds clathrin through the C-terminal regions of the β and γ subunits and functions in clathrin-dependent transport, even though the β subunit also lacks the folded appendage domains of mammalian β subunits (e.g. Yeung and Payne. 2001. Traffic 2(8):565-76). By extension, other fungal AP complexes with β subunits lacking folded appendage domains could have the potential to interact with clathrin through clathrin binding sequences in the non-conserved β "hinge" region and α appendage domain. Furthermore, *S. cerevisiae* AP-2 localizes to endocytic sites that are associated with clathrin (Carrol et al. 2009. Dev Cell 17(4):552-60, 2011; Carroll et al., 2012. Mol Biol Cell 23(4):657-68), making it likely that AP-2 functions with clathrin even though it does not exhibit strong clathrin binding (Yeung et al. 1999. Mol Biol Cell 10(11):3643-59). For these reasons, the authors' interpretation of the phylogenetic analysis in Figure 1 is questionable and it is difficult to predict whether clathrin-independent function described for A. nidulans will be a general feature of AP-2 in other higher fungal species. At a minimum, the manuscript should be rewritten to consider the data from *S. cerevisiae* and moderate any conclusions about other fungi.

Other major comments:

1) Throughout the manuscript, the data often consists of single fluorescence images of cell(s) from an experiment. When at all possible, quantification of multiple cells in multiple images should be carried out to support interpretations based on the images presented, as was done in Figure 2. For example, in the analysis of transporter uptake in Figure 3, the authors conclude that *ap2*∆ does not inhibit internalization (subsection “AP complexes are dispensable for membrane traffic and endocytosis of transporters”, last paragraph), but there is little apparent difference between control and endocytic trigger images of AgtA-GFP or FurA-GFP, suggesting that little internalization occurred in response to the endocytic trigger in these strains. Quantification of internalization or the level of transporter remaining at the cell surface could address this issue. In Figure 7, the authors conclude that the effects of double mutants on morphology is additive (subsection “Further genetic and cellular evidence that AP-2 is involved in apical lipid maintenance”, first paragraph) but most of the double mutant images resemble the single ap2∆ mutant. Some measurement of morphological phenotype is needed to support the conclusion that mutant effects are additive. In addition, Figure 4 would benefit from quantification of morphology and types of septa to allow comparison to ap2∆ mutants in Figure 2; Figure 5 would benefit from quantitation of the ratio of hyphae in each strain with polarized vs non-polarized protein distribution; Figure 8 would benefit from quantitation of polarized distribution of filipin staining.

2) Based on the images in Figure 6, the measured levels of co-localization in some cases are somewhat surprising. For example, whereas the patterns of SagA or SlaB are quite similar to that of AP-2, the same is not true of DnfA, DnfB, and SynA. In these cases, the proteins localize at the apex while AP-2 concentrates at the collar. However, the quantified levels of AP-2 colocalization with these proteins are very similar to those of AP-2 with SagA or SlaB (subsection “AP-2 co-localizes with DnfA, DnfB, SagA and SlaB, but not with clathrin or UapA”, third paragraph). Considering the discordance between the images and quantitation, there is some concern that the resolution/quality of the images, and/or the co-localization methodology is not sufficient to distinguish between actual colocalization and diffuse overlap between proteins localized to distinct but nearby sites. The authors should address this issue and also provide more detail about how the colocalization analysis in Figure 6 was carried out, including how the background was set in each case.

3) The pattern of clathrin fluorescence in Figure 6 is distinct from all other plasma membrane and endocytic proteins analyzed, and probably reflects predominant localization at the Golgi, as is observed in *S. cerevisiae*. There do appear to be weak cortical clathrin puncta at the apex and collar regions, which could represent endocytic clathrin coats. Consequently, it may be more informative to assess colocalization of AP-2 and clathrin at endocytic sites by analyzing higher magnification/resolution images of the apex and collar regions. Alternatively, TIRF microscopy, if possible, might be a more rigorous assay for colocalization of AP-2 with all of the proteins analyzed in Figure 6, including clathrin.

4) The functional significance of the double mutant phenotypes in Figure 7 is not clear. Using deletion mutations, negative genetic interactions could reflect inactivation of functionally distinct pathways or could be due to inactivating two factors in the same pathway that when inactivated alone only partially affect the pathway. The significance of the tested genetic interactions would be more convincing if a mutation in a functionally distinct pathway, such as thiA-claL, does not exhibit genetic interactions with *ap2*∆.

5) The double mutants in Figure 8 are not referred to in the text. Most resemble the single *ap2*∆ mutants, so if the authors have included the double mutants to show additive effects, some quantitation is needed. Also, it is curious that the *ap2*∆ *stoA*∆ double mutant displays polarized filipin staining similar to wild-type, whereas the single mutants exhibit clear defects. The authors should address this phenotype.

*Reviewer #3:*

The manuscript by Martzoukou et al. provides evidence suggesting that in the filamentous yeast *Aspergillus nidulans*: 1) clathrin plays endocytic roles in membrane transporter internalization, which are independent of the AP-2 adaptor; 2) the AP-2 adaptor is required for polarity maintenance and proper hypha growth and; 3) AP-2 plays a role in maintenance of lipid polarity, which is clathrin-independent and which probably implies the clathrin-independent internalization of the lipid flippases DnfA and DnfB. From these three main conclusions, the last one is most novel, provided that AP-2-independent clathrin functions in endocytosis have already been demonstrated in mammalian cells and yeast and that a role of AP-2 in maintenance of polarity has already been demonstrated in *C. albicans* and *S. cerevisiae*. The 3rd conclusion is novel and provides an interesting research avenue but the data supporting it is rather preliminary. The third conclusion is mainly supported by three observations: 1) the sequence known to mediate direct clathrin binding is missing from the β2 adaptin subunit of higher fungi; 2) depletion of the clathrin light chain with a repressible promoter does not cause depolarization of the lipid flippases or ergosterol, whereas depletion of the AP-2 µ or σ subunits does and finally; 3) the AP-2 complex barely co-localizes with a ClaL-GFP construct. There are some concerns regarding all these experiments or the conclusions derived from there. For example, even AP-2 might have lost its capacity to directly bind clathrin in dikarya, it might still be indirectly recruited to clathrin pits and vesicles through binding to other endocytic adaptors. Thus, one would need to demonstrate that AP-2 assembled at the plasma membrane do not colocalize at any time (in time lapse movies) with clathrin. Figure 6 intends to address this issue by showing that the degree of co-localization of AP-2 and the clathrin light chain is lower than the co-localization with other endocytic markers. This could be a valid argument if the authors could assure that the ClaL-GFP construct is competent for its endocytic functions. This construct shows a somehow strange pattern showing big intracellular clumps, while staining at the plasma membrane seems to be completely absent. The author should maybe know that C-terminal tagging of the clathrin light chain in *S. cerevisiae* results in a partially miss-functional protein. Analysis of the co-localization with the endogenous protein or with an N-terminal GFP-ClaL chimera should be included. Also, functional analysis of the GFP and mCherry chimeras used should be provided. Finally, most experiments that show phenotypical differences between the knock out of the AP-2 subunits and clathrin are performed with the constitutive deletions of the AP-2 subunits but with a thiamine repressible construct for the ClaL. It is not clearly defined in the figure legend or the Materials and methods for how long and to what extend the expression of the clathrin light chain is repressed. It might well be that under the conditions used, the ClaL is not fully depleted. Further, the levels of the clathrin heavy chain and its capacity to form triskelions should be analyzed under these conditions. In *S. cerevisiae*, it is known that constitutive deletion of the clathrin light chain cause strong membrane traffic phenotypes because it results in concomitant depletion of the clathrin heavy chain and its inability to form triskelions. However, most *clc1*∆ phenotypes can be overcome by overexpression of the heavy chain (including triskelion assembly). It might well be that under the experimental conditions used in this work, enough triskelions are still present in the cell to support endocytosis of the lipid flippases. To provide strong evidence that the DnfA and DnfB flippases are internalized in an Ap-2 dependent and clathrin-independent manner, the authors would need to analyze the localization of the flippases in the constitutive ClaL mutant and in clathrin heavy chain depleted cells. Further, the authors would need to demonstrate that the flippases can be crosslinked to AP-2 but not to clahtrin. Finally, it would be interesting to investigate the localization of DnfA in a double AP-2-mu and ClaL Knock out. It could be that clathrin deficiency reestablished DnfA polarization by compensating an endocytic defect with slower secretion or recycling rates. If this is the case, depletion of clathrin in an AP-2 mutant would restore polarity. If this is not the case, the double mutant should show a defect similar to the AP-2 mutant.

[Editors' note: further revisions were requested prior to acceptance, as described below.]

Thank you for resubmitting your work entitled "The AP-2 complex has a specialized clathrin-independent role in apical endocytosis and polar growth in fungi" for further consideration at *eLife*. Your revised article has been favorably evaluated by Randy Schekman (Senior and Reviewing editor) and two reviewers.

The manuscript has been improved but there are some remaining issues that need to be addressed before acceptance, as outlined below:

The reviewers would very much like to give the authors another chance to address the weaknesses identified by #3 (full comments below). It should be possible to convincingly show that ClaH is efficiently knocked down in the thiamine experiments that target depletion of ClaH! Quantitative immunoblots should be a good control for that. We are not convinced that endocytic assays such as FM4-64 uptake or LatA treatments are able to discriminate between the clathrin-dependent and the AP-2-dependent pathways, as it is likely that the two share parts of the machinery (It appears that what differs between the two is cargo and localization – both addressed by the authors).

We ask that you do the following:

1) Immunoblots of ClaH in the thiA-ClaH experiments, at the different time points and conditions; +/- thiamine, o/n, 5h and 10h after regrowth).

2) Colocalization of ClaL-RFP with AP2-GFP by TIRF, although, as mentioned before, we do expect a certain degree of colocalization, which does not exclude different functions – but the degree of co- or distinct localization of AP2 and ClaL-RFP on the PM will influence our judgement about the validity of your conclusions.

3) Control wild type data in Figure 3 and Figure 7, as reviewer #3 indicated earlier.

Reviewer #1:

In the revision of the manuscript, the authors have addressed all relevant points raised by the reviewers. As far as I can judge, they have achieved that in a satisfactory manner. In particular, the fluorescence imaging results have been quantified and the authors have obtained more and clearer experimental evidence for distinct roles and localizations of clathrin and AP-2. Thus, I think the manuscript in its current form is well written, and the conclusions made are adequately backed up by data. Most importantly, the major claim that AP-2 has a clathrin-independent role in *A. nidulans* is supported by quantitative experimental data. The manuscript is a very interesting addition to the general recent importance given to clathrin-independent pathways in the endocytosis field, as it emphasizes the variability of endocytic pathways that have evolved from similar sets of protein components.

*Reviewer #3:*

The authors have made an effort to answer several of the points raised by the reviewers but in my opinion, the major concerns have not been addressed satisfactorily. The main point of the manuscript is that AP-2 has a clathrin-independent role maintaining the polarity of DnfA and B localization. In this context, the negative results showing that clathrin does not co-localize with AP-2 and that clathrin does not play a role in the polarization of the lipid flippases are both essential and need to be properly controlled.

Regarding the first point, all reviewers asked for evidence demonstrating the functionality of the ClaL-GFP construct. The authors now add a figure that actually shows that indeed the ClaL-GFP does not fully complement growth. The authors argue that the localization of the ClaL-GFP is fine because the ClaL-GFP pattern resembles somehow the localization of ClaL-mRFP (in terms of showing some clumps in the cytosols) and because the ClaL-mRFP complements the Uap-GFP uptake defect of ClaL mutants. They should then use the ClaL-mRFP to check co-localization with AP-2, not ClaL-GFP (which shows some obvious defects in growth). Even if they use the ClaL-mRFP (that seems to complement better than the ClaL-GFP), most CLaL signal appears in the Golgi and will mask possible co-localization analysis with AP-2 at the plasma membrane. The authors still need to apply TIRF to analyze the co-localization of CLa and AP-2. They only show TIRF to demonstrate that some CLa-GFP seems to be recruited to the plasma membrane. Indeed, in the images shown in Figure 6, one can see a clear co-localization of AP-2 and clathrin in the subapical region, if the putative Golgi signal is subtracted.

With regard to the second point, the authors did not analyzed the extent of depletion of the clathrin heavy and light chains in the CLaL depletion experiments, as requested. This is particularly important given that they cannot used CLaH depletion to show lack of a defect in the DnfA and B localization, since the strain shows major alterations in the secretory pathway upon long thiamine exposure. The experiment depleting CLaH for 5 hours is rather useless because clathrin is a quite stable protein and therefore, most of it, might still be there. The authors would at least need to demonstrate that under the experimental conditions used, other clathrin-dependent functions are significantly affected.

Finally, even though the experiments are maybe not essential to the major point of the article, it is unacceptable that the experiments shown in Figure 3 as well as Figure 7 do not show the wild type control under the same experimental conditions.

---

## [Author Response]

*Essential revisions:*

*1) One of the major findings is presented in Figure 5; the loss of polar localization of the flippases in absence of AP-2, but not in absence of clathrin. Given the importance of these results, they require quantification, and perhaps a higher number of representative example images. Would it be possible to quantify the fluorescence at the tips vs. fluorescence along the hyphae for mutants and wild type cells, for example by measuring the fluorescence intensities along line profiles at defined distances from the tips?*

Both quantification of localization of the flippases (revised Figure 5) and a higher number of representative example images (Figure 5—figure supplement 1) are included in the revised manuscript version. Quantification of the fluorescence at the tips vs. fluorescence along the hyphae was also performed and shown in the revised Figure 5. These quantifications confirm our initial statement that in the absence of AP-2 polar localization of the flippases is lost.

*Throughout the manuscript, the data often consists of single fluorescence images of cell(s) from an experiment. When at all possible, quantification of multiple cells in multiple images should be carried out to support interpretations based on the images presented, as was done in Figure 2. For example, in the analysis of transporter uptake in Figure 3, the authors conclude that ap2∆ does not inhibit internalization (subsection “AP complexes are dispensable for membrane traffic and endocytosis of transporters”, last paragraph), but there is little apparent difference between control and endocytic trigger images of AgtA-GFP or FurA-GFP, suggesting that little internalization occurred in response to the endocytic trigger in these strains. Quantification of internalization or the level of transporter remaining at the cell surface could address this issue.*

We performed quantification of endocytosis for all transporters, by measuring vacuolar GFP fluorescence and vacuolar surface, as shown in revised Figure 3. These measurements confirm that transporter endocytosis operates normally in the absence of AP-2. What might have escaped to the reviewers is that different transporters exhibit different endocytic ‘sensitivities’. For example, FurA (allantoin transporter) is extremely stable in the absence of an endocytic signal and only partially internalized in the presence of endocytic triggers. On the other extreme, FcyB (purine-cytosine transporter) and AgtA (acidic amino acid transporter) are rather unstable, showing a degree of constitutive endocytosis even in the absence of endocytic signals. This difference in stabilities is in fact the reason we selected different transporters for testing the role of AP-2 or clathrin in transporter endocytosis. So, in some cases, as in AgtA-GFP or FurA-GFP, which the reviewer specifically refers, the images, indeed, show little difference. We believe however that the quantifications presented in the revised version of the manuscript clearly dismiss any doubt on the dispensability of AP-2 in transporter endocytosis.

*In Figure 7, the authors conclude that the effects of double mutants on morphology is additive (subsection “Further genetic and cellular evidence that AP-2 is involved in apical lipid maintenance”, first paragraph) but most of the double mutant images resemble the single ap2∆ mutant. Some measurement of morphological phenotype is needed to support the conclusion that mutant effects are additive.*

Quantitative measurements of morphological phenotypes are included in revised Figure 7. These confirm the distinct and rather additive phenotype of the double mutants.

*In addition, Figure 4 would benefit from quantification of morphology and types of septa to allow comparison to ap2∆ mutants in Figure 2.*

Quantitative measurements of morphological phenotypes are included in revised Figure 4. These confirm that lack of ClaL leads to tip swelling, while it does not affect polarity maintenance and normal septum formation, whereas lack of ClaH (see below) also did not affect polarity maintenance and normal septum formation, but led to severe morphological changes of hyphae.

*Figure 5 would benefit from quantitation of the ratio of hyphae in each strain with polarized vs non-polarized protein distribution.*

Quantification added. See above (first part of our answer)

*Figure 8 would benefit from quantification of polarized distribution of filipin staining.*

Quantification of polarized distribution of filipin along the tips is now included in revised Figure 8, supporting our original conclusions.

*2) The reviewers are particularly concerned about the functionality of the ClaL-GFP construct and the extent of the depletion of the clathrin heavy and light chains in the experiments with thiamine. The author should know that C-terminal tagging of the clathrin light chain in S. cerevisiae results in a partially miss-functional protein, so they should provide evidence that ClaL-GFP is a fully functional protein.*

Given the importance of confirming the non-colocalization of AP-2 and clathrin for the present work, we obtained additional evidence to support this finding, basically by showing that GFP- or mRFP-tagged ClaL are functional and that their localization is the expected one for clathrin. The relevant experiments are now highlighted in Figure 6—figure supplement 2. In summary:

Panel A shows that strains expressing the chimeric ClaL-GFP or ClaL-mRFP proteins grow similar to wild-type, contrasting the severe growth defects associated with lack of ClaL expression. Panel B shows that the localization of ClaL-GFP or ClaL-mRFP is identical and compatible with the expected localization for clathrin, as in both cases ClaL marks Golgi-like structures and cortical foci. Panel C confirms, using TIRF microscopy, that a fraction of ClaL is associated with the plasma membrane. Panel D shows a rather weak association of ClaL with subcortical regions of the collar endocytic region. Finally, and most importantly, panel E shows that strains expressing ClaL-mRFP are fully active in respect to UapA-GFP endocytosis, contrasting the block of UapA endocytosis when ClaL is not functional. Last, but not least, Schultzhaus et al., 2016, have shown a very similar subcellular localization of ClaH-GFP, which they further confirm reflects localization principally in the late Golgi, and only a weak association with the sub-apical collar region. In addition, they also show that ClaH and ClaL co-localize significantly. Thus, all evidence shows that clathrin and AP-2 localization is distinct. AP-2 localization is compatible with a protein involved in endocytosis taking place in the collar region, rather than in secretion or exocytosis, where clathrin seems to have an essential role.

*There is the possibility that ClaL depletion does not have a strong effect on triskelia, so they should heavy chain depletion. However, if the heavy chain depletion also does not affect flippase polarization, the clear genetic distinction between the pathways would be sufficient for an initial report.*

In the revised manuscript we have deleted *claH* and also made a strain expressing a regulatable knock-down allele of *claH,* using the *thiA_p_* promoter (*thiAp-claH*). The total deletion of ClaH was lethal, as it was only rescued in heterokaryotic transformants, while the *thiA_p_-claH* stain grew normally in the absence of thiamine, but showed practically no colony growth in the presence of thiamine (Revised Figure 4), or severely modified hyphae morphology under the microscope (Figure 4). These phenotypes were also quantified (Figure 4). Identical results are presented in Schultzhaus et al., 2016b, where the authors used a different promoter to shut-off *claH* transcription, as the total ClaH deletion was also lethal in their case.

We subsequently used the *thiA_p_-claH* strain to study the role ClaH in both UapA endocytosis and in DnfA or DnfB subcellular distribution. Results presented in revised Figure 4 show that ab initio knockdown of ClaH (addition of thiamine from the beginning of growth) results in mislocalization of UapA-GFP, which in this case shows a rather diffuse cytoplasmic appearance, instead of the normal cortical marking of the plasma membrane. This picture was obtained independently from the presence or absence of any endocytic trigger (e.g. ammonium). Given that ClaH is principally localized in late Golgi, as rigorously shown in the recent publication of Schultzhaus et al., 2016, the picture we obtained in respect to the role of ClaH in UapA-GFP localization is compatible with that expected in case of severe block of conventional cargo (transporter) secretion. Interestingly, when ClaH expression is repressed after an initial period of pre-growth (that is, after of UapA-GFP has been secreted to the PM), a significant fraction of UapA-GFP is detected in the PM in the absence of an endocytic trigger, and some is still detectable after the endocytic trigger. This might suggest that ClaH, similar to ClaL, is also needed for transporter endocytosis. The fact that we detect significant turnover of UapA-GFP in the absence of ClaH might well be related to Golgi and endosome collapse and progressive hyphae lethality, which in turn is expected to create a dramatic cellular stress resulting in massive internalization and turnover of plasma membrane transporters/cargoes. Additional evidence for the role of ClaH in conventional secretion via the Golgi was obtained by following the localization of DnfA-GFP or DnfB-GFP in the relevant knockdown strains (revised Figure 5). As in the case of UapA-GFP, when ClaH is repressed from the beginning of growth, DnfA-GFP or DnfB-GFP appear to be cytoplasmically localized, marking bulbous, cytoplasmic bodies and a membrane-like network, compatible with Golgi/endosome collapse. Importantly however, when ClaH expression is repressed after an initial period of pre-growth (that is, after DnfA-GFP or DnfB-GFP have been polarly localized), the pre-established polarity of both flippases is conserved.

A similar picture to the one obtained when ClaH is repressed, concerning the localization of DnfA-GFP or DnfB-GFP, is also obtained when we add Brefeldin A (BFA) in non-repressed conditions (see Figure 5 and Figure 5—figure supplement 2, lower panel). In this case, however, polar localization was re-established after longer periods of exposure to BFA, as expected based in the reversibility of BFA on Golgi functioning, which has been rigorously shown by Pantazopoulou & Peñalva, 2009. The above results strongly suggest that in the absence of a functional ClaH, conventional secretion via the Golgi is blocked so that, among other defects, cargoes (transporters or apical markers) do not reach their destination.

Additionally, our results strongly suggest that the two chains of clathrin have distinct roles in transporter secretion and endocytosis. ClaL proved dispensable for Golgi functioning and transporter secretion towards the plasma membrane, but essential for transporter endocytosis. ClaH was absolutely essential for Golgi functioning and cargo secretion, and probably also for cargo endocytosis. Whether these results are specific for the cargo studied (UapA) or to the endocytic conditions imposed, or whether they reflect the role of clathrin in transporter endocytosis in general, is an issue worth studying, but falls beyond the scope of the present work and will be studied separately.

We strongly believe that the results summarized in Figure 4 and Figure 5 definitely support the idea that AP-2 and clathrin have distinct trafficking and endocytic roles in *A. nidulans,* and probably other filamentous fungi.

*3) Another major concern is that the evolutionary significance of the findings is somewhat uncertain. The authors propose that clathrin-independent function applies to AP-2 in all dikarya (e.g. Abstract), and suggest that the AP-1 complex in dikarya also functions in a clathrin-independent manner (Discussion, third paragraph). These proposals are based primarily on the phylogenetic analysis in Figure 1, which shows that higher fungal species encode β subunits that are shorter than mammalian β subunits, lacking the folded domains in mammalian β appendages that are involved in binding clathrin and clathrin-accessory proteins. However, there is direct evidence that the S. cerevisaie AP-1 complex binds clathrin through the C-terminal regions of the β and γ subunits and functions in clathrin-dependent transport, even though the β subunit also lacks the folded appendage domains of mammalian β subunits (e.g. Yeung and Payne. 2001. Traffic 2(8):565-76). By extension, other fungal AP complexes with β subunits lacking folded appendage domains could have the potential to interact with clathrin through clathrin binding sequences in the non-conserved β "hinge" region and α appendage domain. Furthermore, S. cerevisiae AP-2 localizes to endocytic sites that are associated with clathrin (Carrol et al. 2009. Dev Cell 17(4):552-60, 2011; Carroll et al., 2012. Mol Biol Cell 23(4):657-68), making it likely that AP-2 functions with clathrin even though it does not exhibit strong clathrin binding (Yeung et al. 1999. Mol Biol Cell 10(11):3643-59). For these reasons, the authors' interpretation of the phylogenetic analysis in Figure 1 is questionable and it is difficult to predict whether clathrin-independent function described for A. nidulans will be a general feature of AP-2 in other higher fungal species. At a minimum, the manuscript should be rewritten to consider the data from S. cerevisiae and moderate any conclusions about other fungi.*

We agree with the reviewers that lack of the known clathrin binding domains in AP-2 (or AP-1) does not exclude, a priori, a non-conventional association of AP complexes with clathrin. In the revised manuscript, we moderated our conclusions coming from the phylogenetic analysis and added missing references, as proposed by the reviewers. However, subsequent experimental evidence supported the non-interaction of AP-2 with clathrin. More specifically, we showed that ClaL and AP-2 do not co-localize and, most importantly, the corresponding null mutants have distinct cellular and molecular phenotypes. We have not performed co-localization studies of ClaH with AP-2, but Schultzhaus et al., 2016b have shown that ClaH-GFP labels mostly the late Golgi and only weakly the collar. Such localization is clearly distinct from the one obtained with AP-2, which labels principally the collar region and not at all the Golgi. In Schultzhaus et al., 2016b, it is also shown that ClaH and ClaL co-localize significantly (although not totally), providing further indirect evidence that AP-2 does not co-localize with clathrin.

*4) A key indication for a spatial uncoupling of clathrin- and AP-2-dependent endocytosis is the colocalization data shown in Figure 6. ClCl^-^GFP (clathrin) does not appear to localize at the plasma membrane. What is seen in the micrograph is probably clathrin on intracellular membranes, which is likely to obscure weaker possible patches on the plasma membrane. However, to support the conclusions made, it is crucial to see where on the plasma membrane clathrin performs its endocytic function. Given that it is involved in transporter uptake like SlaB, one would expect, besides the expected intracellular localization, similar PM-patches for ClcL as for SlaB. Could the authors try to overcome this problem by using different imaging techniques, for example by TIRF microscopy, to visualize the plasma membrane only rather than the intracellular membranes? If not, it may be sufficient to assess colocalization of AP-2 and clathrin at endocytic sites by analyzing higher magnification/resolution images of the apex and collar regions.*

In the revised manuscript we have overcome this issue by using TIRF microscopy, which confirmed that a fraction of ClaL-GFP very probably associates with the PM, but also by showing higher magnification images of the apex and collar regions (revised Figure 6—figure supplement 2). The weak association of ClaL with the PM or the collar region was also very similar to the results obtained with ClaH in Schultzhaus et al., 2016b, where it is also concluded that clathrin has a minor role in apical endocytosis.

*The full comments of the reviewers follow. However, only the essential revisions listed above will be required for your revised submission. Additional major questions below may be addressed at your discretion.*

*Reviewer #2:*

*[…] Other major comments:*

*1) Throughout the manuscript, the data often consists of single fluorescence images of cell(s) from an experiment. When at all possible, quantification of multiple cells in multiple images should be carried out to support interpretations based on the images presented, as was done in Figure 2. For example, in the analysis of transporter uptake in Figure 3, the authors conclude that ap2∆ does not inhibit internalization (subsection “AP complexes are dispensable for membrane traffic and endocytosis of transporters”, last paragraph), but there is little apparent difference between control and endocytic trigger images of AgtA-GFP or FurA-GFP, suggesting that little internalization occurred in response to the endocytic trigger in these strains. Quantification of internalization or the level of transporter remaining at the cell surface could address this issue. In Figure 7, the authors conclude that the effects of double mutants on morphology is additive (subsection “Further genetic and cellular evidence that AP-2 is involved in apical lipid maintenance”, first paragraph) but most of the double mutant images resemble the single ap2∆ mutant. Some measurement of morphological phenotype is needed to support the conclusion that mutant effects are additive. In addition, Figure 4 would benefit from quantification of morphology and types of septa to allow comparison to ap2∆ mutants in Figure 2; Figure 5 would benefit from quantitation of the ratio of hyphae in each strain with polarized vs non-polarized protein distribution; Figure 8 would benefit from quantitation of polarized distribution of filipin staining.*

Addressed as described earlier.

*2) Based on the images in Figure 6, the measured levels of co-localization in some cases are somewhat surprising. For example, whereas the patterns of SagA or SlaB are quite similar to that of AP-2, the same is not true of DnfA, DnfB, and SynA. In these cases, the proteins localize at the apex while AP-2 concentrates at the collar. However, the quantified levels of AP-2 colocalization with these proteins are very similar to those of AP-2 with SagA or SlaB (subsection “AP-2 co-localizes with DnfA, DnfB, SagA and SlaB, but not with clathrin or UapA”, third paragraph). Considering the discordance between the images and quantitation, there is some concern that the resolution/quality of the images, and/or the co-localization methodology is not sufficient to distinguish between actual colocalization and diffuse overlap between proteins localized to distinct but nearby sites. The authors should address this issue and also provide more detail about how the colocalization analysis in Figure 6 was carried out, including how the background was set in each case.*

In the lower panel of Figure 6 we show that AP-2 does not co-localize significantly with DnfA and SynA when the quantification is restricted to the tip, as expected for a factor involved exclusively in endocytosis. When co-localization measurements reflect the broader area of the apical and sub-apical region, where both exocytosis and endocytosis operate (upper panel in 6C), co-localization of AP-2 with recycling cargoes (SynA and DfnA) is observed. For more details on co-localization see Materials and methods (use of image J and ICY image analysis).

*3) The pattern of clathrin fluorescence in Figure 6 is distinct from all other plasma membrane and endocytic proteins analyzed, and probably reflects predominant localization at the Golgi, as is observed in S. cerevisiae. There do appear to be weak cortical clathrin puncta at the apex and collar regions, which could represent endocytic clathrin coats. Consequently, it may be more informative to assess colocalization of AP-2 and clathrin at endocytic sites by analyzing higher magnification/resolution images of the apex and collar regions. Alternatively, TIRF microscopy, if possible, might be a more rigorous assay for colocalization of AP-2 with all of the proteins analyzed in Figure 6, including clathrin.*

Addressed as described earlier.

*4) The functional significance of the double mutant phenotypes in Figure 7 is not clear. Using deletion mutations, negative genetic interactions could reflect inactivation of functionally distinct pathways or could be due to inactivating two factors in the same pathway that when inactivated alone only partially affect the pathway. The significance of the tested genetic interactions would be more convincing if a mutation in a functionally distinct pathway, such as thiA-claL, does not exhibit genetic interactions with ap2∆.*

Interesting point to be tested in the future.

*5) The double mutants in Figure 8 are not referred to in the text. Most resemble the single ap2∆ mutants, so if the authors have included the double mutants to show additive effects, some quantitation is needed. Also, it is curious that the ap2∆ stoA∆ double mutant displays polarized filipin staining similar to wild-type, whereas the single mutants exhibit clear defects. The authors should address this phenotype.*

We removed the double mutants as they give complex phenotypes difficult to rationalize at present, so we think that they do not contribute significantly to our conclusions.

*Reviewer #3:*

*The manuscript by Martzoukou et al. provides evidence suggesting that in the filamentous yeast Aspergillus nidulans: 1) clathrin plays endocytic roles in membrane transporter internalization, which are independent of the AP-2 adaptor; 2) the AP-2 adaptor is required for polarity maintenance and proper hypha growth and; 3) AP-2 plays a role in maintenance of lipid polarity, which is clathrin-independent and which probably implies the clathrin-independent internalization of the lipid flippases DnfA and DnfB. From these three main conclusions, the last one is most novel, provided that AP-2-independent clathrin functions in endocytosis have already been demonstrated in mammalian cells and yeast and that a role of AP-2 in maintenance of polarity has already been demonstrated in C. albicans and S. cerevisiae. The 3rd conclusion is novel and provides an interesting research avenue but the data supporting it is rather preliminary. The third conclusion is mainly supported by three observations: 1) the sequence known to mediate direct clathrin binding is missing from the β2 adaptin subunit of higher fungi; 2) depletion of the clathrin light chain with a repressible promoter does not cause depolarization of the lipid flippases or ergosterol, whereas depletion of the AP-2 µ or σ subunits does and finally; 3) the AP-2 complex barely co-localizes with a ClaL-GFP construct. There are some concerns regarding all these experiments or the conclusions derived from there. For example, even AP-2 might have lost its capacity to directly bind clathrin in dikarya, it might still be indirectly recruited to clathrin pits and vesicles through binding to other endocytic adaptors. Thus, one would need to demonstrate that AP-2 assembled at the plasma membrane do not colocalize at any time (in time lapse movies) with clathrin. Figure 6 intends to address this issue by showing that the degree of co-localization of AP-2 and the clathrin light chain is lower than the co-localization with other endocytic markers. This could be a valid argument if the authors could assure that the ClaL-GFP construct is competent for its endocytic functions. This construct shows a somehow strange pattern showing big intracellular clumps, while staining at the plasma membrane seems to be completely absent. The author should maybe know that C-terminal tagging of the clathrin light chain in S. cerevisiae results in a partially miss-functional protein. Analysis of the co-localization with the endogenous protein or with an N-terminal GFP-ClaL chimera should be included. Also, functional analysis of the GFP and mCherry chimeras used should be provided. Finally, most experiments that show phenotypical differences between the knock out of the AP-2 subunits and clathrin are performed with the constitutive deletions of the AP-2 subunits but with a thiamine repressible construct for the ClaL. It is not clearly defined in the figure legend or the Materials and methods for how long and to what extend the expression of the clathrin light chain is repressed. It might well be that under the conditions used, the ClaL is not fully depleted. Further, the levels of the clathrin heavy chain and its capacity to form triskelions should be analyzed under these conditions. In S. cerevisiae, it is known that constitutive deletion of the clathrin light chain cause strong membrane traffic phenotypes because it results in concomitant depletion of the clathrin heavy chain and its inability to form triskelions. However, most clc1∆ phenotypes can be overcome by overexpression of the heavy chain (including triskelion assembly). It might well be that under the experimental conditions used in this work, enough triskelions are still present in the cell to support endocytosis of the lipid flippases. To provide strong evidence that the DnfA and DnfB flippases are internalized in an Ap-2 dependent and clathrin-independent manner, the authors would need to analyze the localization of the flippases in the constitutive ClaL mutant and in clathrin heavy chain depleted cells. Further, the authors would need to demonstrate that the flippases can be crosslinked to AP-2 but not to clahtrin. Finally, it would be interesting to investigate the localization of DnfA in a double AP-2-mu and ClaL Knock out. It could be that clathrin deficiency reestablished DnfA polarization by compensating an endocytic defect with slower secretion or recycling rates. If this is the case, depletion of clathrin in an AP-2 mutant would restore polarity. If this is not the case, the double mutant should show a defect similar to the AP-2 mutant.*

Major points requiring revision were addressed as described earlier. We did not perform cross-link experiments though. We tried to use BiFC assays to show interactions of AP-2 and DnfA, but these were not successful. We did not investigate the localization of DnfA in a double AP-2 and ClaL mutant, but this seems an interesting proposal for the future. The present work does not intend to study possible interactions of different trafficking pathways in specific mutant backgrounds, but rather present strong evidence on the distinct roles of AP-2 and clathrin. Finally, the thiA_p_ promoter is a very well-studied tool in Aspergillus laboratories and there is evidence that its repression by 5-10 mM thiamine for more than 3 h leads to depletion of gene products expressed through it. In our case we used both 5 and 10 h repression with identical results. Repression of ClaL or ClaH for 5h proved to fully block transporter endocytosis or to lead Golgi disfunctioning similar to the one we get using BFA, strongly suggesting the in our conditions clathrin heavy chains are totally depleted.

[Editors' note: further revisions were requested prior to acceptance, as described below.]

*The manuscript has been improved but there are some remaining issues that need to be addressed before acceptance, as outlined below:*

*The reviewers would very much like to give the authors another chance to address the weaknesses identified by #3 (full comments below). It should be possible to convincingly show that ClaH is efficiently knocked down in the thiamine experiments that target depletion of ClaH! Quantitative immunoblots should be a good control for that. We are not convinced that endocytic assays such as FM4-64 uptake or LatA treatments are able to discriminate between the clathrin-dependent and the AP-2-dependent pathways, as it is likely that the two share parts of the machinery (It appears that what differs between the two is cargo and localization – both addressed by the authors).*

*We ask that you do the following:*

*1) Immunoblots of ClaH in the thiA-ClaH experiments, at the different time points and conditions; +/- thiamine, o/n, 5h and 10h after regrowth).*

The immunoblots concerning ClaH were performed and quantified as asked. ClaH and levels are dramatically reduced (7% of the non-repressed levels) after 10 h or o/n repression by thiamine (practically to the level obtained by continuous presence of thiamine), but not when repression was only for 5 h. A similar result was obtained with ClaL (not shown). Thus, in the revised version we removed data with 5 h repression, and only show results from the epifluorescence analysis of samples repressed for either o/n (for ClaL) or 10 h (for ClaH). Repression and depletion of ClaH led to an apparent dramatic block of cargo secretion, similar to the block obtained after o/n repression, or BFA addition. Although polar deposition of DnfA or DnfB was significantly lost when ClaH was depleted, a degree of polarization was still evident. The effect of ClaH depletion on the polar localization of flippases is similar to the effect of BFA, strongly suggesting that a block in secretion is essential for *maintaining*, rather than *establishing,* the polar localization of apical markers. The non-essentiality of the clathrin complex for the initial polar depositioning of DnfA or DnfB is otherwise clearly supported in samples where ClaL is genetically depleted. In this case, where secretion is not blocked, we detected propter polarization of flippases.

We cannot, in principle, formally exclude that DnfA or DnfB polar deposition, despite being fully independent of ClaL, is somehow dependent of ClaH. However, still the more logical scenario is that the observed modification on DnfA or DnfB polar depositioning is a direct consequence of blocked secretion in the absence of ClaH, and as also seen in the presence of BFA. In anyway, this does not affect our primary conclusion that AP-2 and clathrin have distinct roles in secretion (AP-2 independent, ClaH-dependent) and polar maintenance of apical markers (AP-2-dependent, ClaL-independent).

These results, together with additional evidence on the non-colocalization of AP-2 and ClaL in the PM shown by TIRF (see below), as well as, the very distinct phenotypes of AP-2 and clathrin mutants, strongly support our main conclusion on the distinct roles of clathrin and AP2 in cargo traffic.

*2) Colocalization of ClaL-RFP with AP2-GFP by TIRF, although, as mentioned before, we do expect a certain degree of colocalization, which does not exclude different functions – but the degree of co- or distinct localization of AP2 and ClaL-RFP on the PM will influence our judgement about the validity of your conclusions.*

We analyzed the localization of Ap2^σ^-GFP and ClaL-mRFP by TIRF microscopy followed by relevant quantification analysis (see new Figure 7). Practically no co-localization of the two polypeptides was observed (PPC=0.12) in the PM. Only a minor fraction of the two polypeptides co-localized (PPC = 0.34) intracellularly. The P-values are P<0.0001 and P<0.05 for co-localization of AP-2 with ClaL by epifluorescence microscopy and by TIRF, respectively. These data are in perfect line with the rest of our findings, confirming that the function of AP-2 in apical endocytosis is clathrin-independent.

*3) Control wild type data in Figure 3 and Figure 7, as reviewer #3 indicated earlier.*

In the revised manuscript we include the experiments asked concerning the endocytosis of transporters in a wild-type background, and also quantified the degree of endocytic turnover in all cases. We also show the controls asked in Figure 8 for Ap2-GFP (25, 42 o C) (previously Figure 7). The results shown are in perfect line with our original conclusions (as expected given that the control experiments have also been performed independently many times in our laboratory before) concerning the non-essentiality of AP2 in transporter endocytosis.